# FACMAC: Factored Multi-Agent Centralised Policy Gradients

**Bei Peng**[*][†]
University of Liverpool

**Tabish Rashid**[*]
University of Oxford

**Christian A. Schroeder de Witt**[*]
University of Oxford

**Pierre-Alexandre Kamienny**[†]
Facebook AI Research

**Philip H. S. Torr**
University of Oxford

**Wendelin Böhmer**
Delft University of Technology

**Shimon Whiteson**
University of Oxford

## Abstract

We propose **FAC**tored **M**ulti-**A**gent **C**entralised policy gradients (FACMAC), a new method for cooperative multi-agent reinforcement learning in both discrete and continuous action spaces. Like MADDPG, a popular multi-agent actor-critic method, our approach uses deep deterministic policy gradients to learn policies. However, FACMAC learns a centralised but factored critic, which combines per-agent utilities into the joint action-value function via a non-linear monotonic function, as in QMIX, a popular multi-agent $Q$-learning algorithm. However, unlike QMIX, there are no inherent constraints on factoring the critic. We thus also employ a nonmonotonic factorisation and empirically demonstrate that its increased representational capacity allows it to solve some tasks that cannot be solved with monolithic, or monotonically factored critics. In addition, FACMAC uses a centralised policy gradient estimator that optimises over the entire joint action space, rather than optimising over each agent's action space separately as in MADDPG. This allows for more coordinated policy changes and fully reaps the benefits of a centralised critic. We evaluate FACMAC on variants of the multi-agent particle environments, a novel multi-agent MuJoCo benchmark, and a challenging set of StarCraft II micromanagement tasks. Empirical results demonstrate FACMAC's superior performance over MADDPG and other baselines on all three domains.

## 1 Introduction

Significant progress has been made in cooperative multi-agent reinforcement learning (MARL) under the paradigm of *centralised training with decentralised execution* (CTDE) [27, 18] in recent years, both in value-based [41, 31, 37, 32, 44, 30] and actor-critic [23, 9, 36, 13, 8] approaches. Most popular multi-agent actor-critic methods such as COMA [9] and MADDPG [23] learn a *centralised critic* with decentralised actors. The critic is centralised to make use of all available information (i.e., it can condition on the global state and the joint action) to estimate the joint action-value function $Q_{tot}$, unlike a *decentralised critic* that estimates the local action-value function $Q_a$ based only on individual observations and actions for each agent $a$.[3] Even though the joint action-value function

---

[*]Equal contribution. Correspondence to: Bei Peng <bei.peng@liverpool.ac.uk>

[†]Work done while the authors were at the University of Oxford.

[3]COMA learns a single centralised critic for all cooperative agents due to parameter sharing. For each agent the critic has different inputs and can thus output different values for the same state and joint action.

35th Conference on Neural Information Processing Systems (NeurIPS 2021).

these actor-critic methods can represent is not restricted, in practice they significantly underperform value-based methods like QMIX [31] on the challenging StarCraft Multi-Agent Challenge (SMAC) [35] benchmark [33, 32].

In this paper, we propose a novel approach called ***FACtored Multi-Agent Centralised policy gradients*** (FACMAC), which works for both discrete and continuous cooperative multi-agent tasks. Like MADDPG, our approach uses deep deterministic policy gradients [20] to learn decentralised policies. However, FACMAC learns a single *centralised but factored critic*, which factors the joint action-value function $Q_{tot}$ into per-agent utilities $Q_a$ that are combined via a non-linear monotonic function, as in the popular $Q$-learning algorithm QMIX [31]. While the critic used in COMA and MADDPG is also centralised, it is monolithic rather than factored.[4] Compared to learning a monolithic critic, our factored critic can potentially scale better to tasks with a larger number of agents and/or actions. In addition, in contrast to other value-based approaches such as QMIX, there are no inherent constraints on factoring the critic. This allows us to employ rich value factorisations, including *nonmonotonic* ones, that value-based methods cannot directly use without forfeiting decentralisability or introducing other significant algorithmic changes. We thus also employ a nonmonotonic factorisation and empirically demonstrate that its increased representational capacity allows it to solve some tasks that cannot be solved with monolithic, or monotonically factored critics.

In MADDPG, a separate policy gradient is derived for each agent individually, which optimises its policy assuming all other agents' actions are fixed. This could cause the agents to converge to sub-optimal policies in which no single agent wishes to change its action unilaterally. In FACMAC, we use a new *centralised* gradient estimator that optimises over the entire joint action space, rather than optimising over each agent's action space separately as in MADDPG. The agents' policies are thus trained as a single joint-action policy, which can enable learning of more coordinated behaviour, as well as the ability to escape sub-optimal solutions. The centralised gradient estimator fully reaps the benefits of learning a centralised critic, by not implicitly marginalising over the actions of the other agents in the policy-gradient update. The gradient estimator used in MADDPG is also known to be vulnerable to relative overgeneralisation [47]. To overcome this issue, in our centralised gradient estimator, we sample all actions from all agents' current policies when evaluating the joint action-value function. We empirically show that MADDPG can quickly get stuck in local optima in a simple continuous matrix game, whereas our centralised gradient estimator finds the optimal policy. While Lyu et al. [24] recently show that merely using a centralised critic (with per-agent gradients that optimise over each agent's actions separately) does not necessarily lead to better coordination between agents, our centralised gradient estimator re-establishes the value of using centralised critics.

Most recent works on continuous MARL focus on evaluating their algorithms on the multi-agent particle environments [23], which feature a simple two-dimensional world with some basic simulated physics. To demonstrate FACMAC's scalability to more complex continuous domains and to stimulate more progress in continuous MARL, we introduce *Multi-Agent MuJoCo* (MAMuJoCo), a new, comprehensive benchmark suite that allows the study of decentralised continuous control. Based on the popular single-agent MuJoCo benchmark [5], MAMuJoCo features a wide variety of novel robotic control tasks in which multiple agents within a single robot have to solve a task cooperatively.

We evaluate FACMAC on variants of the multi-agent particle environments [23] and our novel MAMuJoCo benchmark, which both feature continuous action spaces, and the challenging SMAC benchmark [35], which features discrete action spaces. Empirical results demonstrate FACMAC's superior performance over MADDPG and other baselines on all three domains. In particular, FACMAC scales better when the number of agents (and/or actions) and the complexity of the task increases. Results on SMAC show that FACMAC significantly outperforms stochastic DOP [46], which recently claimed to be the first multi-agent actor-critic method to outperform state-of-the-art valued-based methods on SMAC, in all scenarios we tested. Moreover, our ablations and additional experiments demonstrate the advantages of both factoring the critic and using our centralised gradient estimator. We show that, compared to learning a monolithic critic, learning a factored critic can: 1) better take advantage of the centralised gradient estimator to optimise the agent policies when the

---

In MADDPG, each agent learns its own centralised critic, as it is designed for general multi-agent learning problems, including cooperative, competitive, and mixed settings.

[4]We use "centralised and monolithic critic" and "monolithic critic" interchangeably to refer to the centralised critic used in COMA and MADDPG, and "centralised but factored critic" and "factored critic" interchangeably to refer to the critic used in our approach.

number of agents and/or actions is large, and 2) leverage a nonmonotonic factorisation to solve tasks that cannot be solved with monolithic or monotonically factored critics.

## 2 Background

We consider a *fully cooperative multi-agent task* in which a team of agents interacts with the same environment to achieve some common goal. It can be modeled as a *decentralised partially observable Markov decision process* (Dec-POMDP) [28] consisting of a tuple $G = \langle \mathcal{N}, S, U, P, r, \Omega, O, \gamma \rangle$. Here $\mathcal{N} \equiv \{1, \ldots, n\}$ denotes the finite set of agents and $s \in S$ describes the true state of the environment. At each time step, each agent $a \in \mathcal{N}$ selects a discrete or continuous action $u_a \in U$, forming a joint action $\mathbf{u} \in \mathbf{U} \equiv U^n$. This results in a transition to the next state $s'$ according to the state transition function $P(s'|s, \mathbf{u}) : S \times \mathbf{U} \times S \rightarrow [0, 1]$ and a team reward $r(s, \mathbf{u})$. $\gamma \in [0, 1)$ is a discount factor. Due to the *partial observability*, each agent $a \in \mathcal{N}$ draws an individual partial observation $o_a \in \Omega$ from the observation kernel $O(s, a)$. Each agent learns a stochastic policy $\pi_a(u_a|\tau_a)$ or a deterministic policy $\mu_a(\tau_a)$, conditioned only on its local action-observation history $\tau_a \in T \equiv (\Omega \times U)^*$. The joint stochastic policy $\boldsymbol{\pi}$ induces a joint *action-value function*: $Q^{\boldsymbol{\pi}}(s_t, \mathbf{u}_t) = \mathbb{E}_{s_{t+1:\infty}, \mathbf{u}_{t+1:\infty}} [R_t | s_t, \mathbf{u}_t]$, where $R_t = \sum_{i=0}^{\infty} \gamma^i r_{t+i}$ is the discounted return. Similarly, the joint deterministic policy $\boldsymbol{\mu}$ induces a joint action-value function denoted $Q^{\boldsymbol{\mu}}(s_t, \mathbf{u}_t)$. We adopt the *centralised training with decentralised execution* (CTDE) paradigm [27, 18], where policy training can exploit extra global information that might be available and has the freedom to share information between agents during training. However, during execution, each agent must act with only access to its own action-observation history.

**VDN and QMIX.** VDN [41] and QMIX [31] are $Q$-learning algorithms for cooperative MARL tasks with discrete actions. They both aim to efficiently learn a centralised but factored action-value function $Q^{\boldsymbol{\pi}}_{tot}$, using CTDE. To ensure consistency between the centralised and decentralised policies, VDN and QMIX factor $Q^{\boldsymbol{\pi}}_{tot}$ assuming additivity and monotonicity, respectively. More specifically, VDN factors $Q^{\boldsymbol{\pi}}_{tot}$ into a sum of the per-agent utilities: $Q^{\boldsymbol{\pi}}_{tot}(\boldsymbol{\tau}, \mathbf{u}; \boldsymbol{\phi}) = \sum_{a=1}^{n} Q^{\pi_a}_a(\tau_a, u_a; \phi_a)$. QMIX, however, represents $Q^{\boldsymbol{\pi}}_{tot}$ as a continuous monotonic mixing function of each agent's utilities: $Q^{\boldsymbol{\pi}}_{tot}(\boldsymbol{\tau}, \mathbf{u}, s; \boldsymbol{\phi}, \psi) = f_\psi(s, Q^{\pi_1}_1(\tau_1, u_1; \phi_1), \ldots, Q^{\pi_n}_n(\tau_n, u_n; \phi_n))$, where $\frac{\partial f_\psi}{\partial Q_a} \geq 0, \forall a \in \mathcal{N}$. This is sufficient to ensure that the global $\arg\max$ performed on $Q^{\boldsymbol{\pi}}_{tot}$ yields the same result as a set of individual $\arg\max$ performed on each $Q^{\pi_a}_a$. Here $f_\psi$ is approximated by a monotonic mixing network, parameterised by $\psi$. Monotonicity can be guaranteed by non-negative mixing weights. These weights are generated by separate *hypernetworks* [12], which condition on the full state $s$. QMIX is trained end-to-end to minimise the following loss:

$$\mathcal{L}(\boldsymbol{\phi}, \psi) = \mathbb{E}_{\mathcal{D}} \left[ \left( y^{tot} - Q^{\boldsymbol{\pi}}_{tot}(\boldsymbol{\tau}, \mathbf{u}, s; \boldsymbol{\phi}, \psi) \right)^2 \right], \tag{1}$$

where the bootstrapping target $y^{tot} = r + \gamma \max_{\mathbf{u}'} Q^{\boldsymbol{\pi}}_{tot}(\boldsymbol{\tau}', \mathbf{u}', s'; \boldsymbol{\phi}^-, \psi^-)$. Here $r$ is the global reward, and $\boldsymbol{\phi}^-$ and $\psi^-$ are parameters of the target $Q$ and mixing network, respectively, as in DQN [26]. The expectation is estimated with a minibatch of transitions sampled from an experience replay buffer $\mathcal{D}$ [21]. During execution, each agent selects actions greedily with respect to its own $Q^{\pi_a}_a$.

**MADDPG.** MADDPG [23] is an extension of DDPG [20] to multi-agent settings. It is an actor-critic, off-policy method that uses the paradigm of CTDE to learn deterministic policies in continuous action spaces. In MADDPG, a separate actor and critic is learned for each agent, such that each agent can have its own arbitrary reward function. It is therefore applicable to either cooperative, competitive, or mixed settings. We assume each agent $a$ has a deterministic policy $\mu_a(\tau_a; \theta_a)$, parameterised by $\theta_a$ (abbreviated as $\mu_a$), and let $\boldsymbol{\mu} = \{\mu_a(\tau_a; \theta_a)\}_{a=1}^{n}$ be the set of all agent policies. In MADDPG, a *centralised and monolithic critic* that estimates the joint action-value function $Q^{\boldsymbol{\mu}}_a(s, u_1, \ldots, u_n; \phi_a)$ is learned for each agent $a$ separately. The critic is said to be *centralised* as it utilises information only available to it during the *centralised* training phase, the global state $s^5$ and the actions of all agents, $u_1, \ldots, u_n$, to estimate the joint action-value function $Q^{\boldsymbol{\mu}}_a$, which is parameterised by $\phi_a$. This joint action-value function is trained by minimising the following loss:

$$\mathcal{L}(\phi_a) = \mathbb{E}_{\mathcal{D}} \left[ \left( y^a - Q^{\boldsymbol{\mu}}_a(s, u_1, \ldots, u_n; \phi_a) \right)^2 \right], \tag{2}$$

---

[5] If the global state $s$ is not available, the centralised and monolithic critic can condition on the joint observations or action-observation histories.

where $y^a = r_a + \gamma Q_a^{\boldsymbol{\mu}}(s', u_1', \ldots, u_n')|_{u_a' = \mu_a(\tau_a'; \theta_a^-)}; \phi_a^-)$. Here $r_a$ is the reward received by each agent $a$, $u_1', \ldots, u_n'$ is the set of target policies with delayed parameters $\theta_a^-$, and $\phi_a^-$ are the parameters of the target critic. The replay buffer $\mathcal{D}$ contains the transition tuples $(s, s', u_1, \ldots, u_n, r_1, \ldots, r_n)$.

The following policy gradient can be calculated individually to update the policy of each agent $a$:

$$\nabla_{\theta_a} J(\mu_a) = \mathbb{E}_{\mathcal{D}}\Big[\nabla_{\theta_a}\mu_a(\tau_a)\nabla_{u_a} Q_a^{\boldsymbol{\mu}}(s, u_1, \ldots, u_n)\big|_{u_a = \mu_a(\tau_a)}\Big], \tag{3}$$

where the current agent $a$'s action $u_a$ is sampled from its current policy $\mu_a$ when evaluating the joint action-value function $Q_a^{\boldsymbol{\mu}}$, while all other agents' actions are sampled from the replay buffer $\mathcal{D}$.

## 3 FACMAC

In this section, we propose a new approach called ***FAC**tored **M**ulti-**A**gent **C**entralised policy gradients* (FACMAC) that uses a centralised but factored critic and a centralised gradient estimator to learn continuous cooperative tasks. We start by describing the idea of learning a centralised but factored critic. We then discuss our new centralised gradient estimator and demonstrate its benefit in a simple continuous matrix game. Finally, we discuss how we adapt our method to discrete cooperative tasks.

### 3.1 Learning a Centralised but Factored Critic

Learning a centralised and monolithic critic conditioning on the global state and the joint action can be difficult and/or impractical when the number of agents and/or actions is large [13]. We thus employ value function factorisation in the multi-agent actor-critic framework to enable scalable learning of a centralised critic in Dec-POMDPs. Another key advantage of adopting value factorisation in an actor-critic framework is that, compared to value-based methods, it allows for a more flexible factorisation as the critic's design is not constrained. One can employ any type of factorisation, including nonmonotonic factorisations that value-based methods cannot directly use without forfeiting decentralisability or introducing other significant algorithmic changes.

Specifically, in FACMAC, all agents share a centralised critic $Q_{tot}^{\boldsymbol{\mu}}$ that is factored as:

$$Q_{tot}^{\boldsymbol{\mu}}(\boldsymbol{\tau}, \mathbf{u}, s; \boldsymbol{\phi}, \psi) = g_\psi\big(s, \{Q_a^{\mu_a}(\tau_a, u_a; \phi_a)\}_{a=1}^n\big), \tag{4}$$

where $\boldsymbol{\phi}$ and $\phi_a$ are parameters of the joint action-value function $Q_{tot}^{\boldsymbol{\mu}}$ and agent-wise utilities $Q_a^{\mu_a}$, respectively. In our canonical implementation which we refer to as FACMAC, $g_\psi$ is a non-linear monotonic function parametrised as a mixing network with parameters $\psi$, as in QMIX [31]. To evaluate the policy, the centralised but factored critic is trained by minimising the following loss:

$$\mathcal{L}(\boldsymbol{\phi}, \psi) = \mathbb{E}_{\mathcal{D}}\Big[\big(y^{tot} - Q_{tot}^{\boldsymbol{\mu}}(\boldsymbol{\tau}, \mathbf{u}, s; \boldsymbol{\phi}, \psi)\big)^2\Big], \tag{5}$$

where $y^{tot} = r + \gamma Q_{tot}^{\boldsymbol{\mu}}(\boldsymbol{\tau}', \boldsymbol{\mu}(\boldsymbol{\tau}'; \boldsymbol{\theta}^-), s'; \boldsymbol{\phi}^-, \psi^-)$. Here $\mathcal{D}$ is the replay buffer, and $\boldsymbol{\theta}^-$, $\boldsymbol{\phi}^-$, and $\psi^-$ are parameters of the target actors, critic, and mixing network, respectively.

Leveraging the flexibility of our approach, namely the lack of restrictions on the form of the critic, we also explore a new nonmonotonic factorisation with full representational capacity. The joint action-value function $Q_{tot}^{\boldsymbol{\mu}}$ is represented as a non-linear nonmonotonic mixing function of per-agent utilities $Q_a^{\mu_a}$. This nonmonotonic mixing function is parameterised as a mixing network, with a similar architecture to $g_\psi$ in FACMAC, but without the constraint of monotonicity enforced by using non-negative weights. We refer to this method as FACMAC-nonmonotonic. Additionally, to better understand the advantages of factoring a centralised critic, we also explore two additional simpler factorisation schemes. These include factoring the centralised critic $Q_{tot}^{\boldsymbol{\mu}}$ into a sum of per-agent utilities $Q_a^{\mu_a}$ as in VDN (FACMAC-vdn), and as a sum of $Q_a^{\mu_a}$ and a state-dependent bias (FACMAC-vdn-s). Our value factorisation technique is general and can be readily applied to any multi-agent actor-critic algorithms that learn centralised and monolithic critics [23, 9, 8].

### 3.2 Centralised Policy Gradients

To update the decentralised policy of each agent, a naive adaptation of the deterministic policy gradient used by MADDPG (shown in (3)) is

$$\nabla_{\theta_a} J(\mu_a) = \mathbb{E}_{\mathcal{D}}\Big[\nabla_{\theta_a}\mu_a(\tau_a)\nabla_{u_a} Q_{tot}^{\boldsymbol{\mu}}(\boldsymbol{\tau}, u_1, \ldots, u_n, s)\big|_{u_a = \mu_a(\tau_a)}\Big]. \tag{6}$$

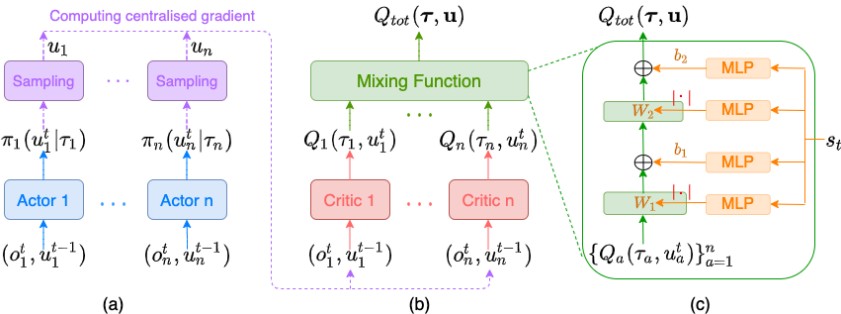

Figure 1: The overall FACMAC architecture. (a) The decentralised policy networks. There is a sampling step since we sample from the categorical distribution when using discrete actions. (b) The centralised but factored critic. (c) The non-linear monotonic mixing function.

Compared to the policy gradient used in MADDPG, the updates of all agents' individual deterministic policies now depend on the single shared factored critic $Q_{tot}^{\boldsymbol{\mu}}$, as opposed to learning and utilising a monolithic critic $Q_a^{\boldsymbol{\mu}}$ for each agent. However, there are two main problems in both policy gradients. First, each agent optimises its own policy assuming all other agents' actions are fixed, which could cause the agents to converge to sub-optimal policies in which no single agent wishes to change its action unilaterally. Second, both policy gradients make the corresponding methods vulnerable to relative overgeneralisation [47] as, when agent $a$ ascends the policy gradient based on $Q_a^{\boldsymbol{\mu}}$ or $Q_{tot}^{\boldsymbol{\mu}}$, only its own action $u_a$ is sampled from its current policy $\mu_a$, while all other agents' actions are sampled from the replay buffer $\mathcal{D}$. The other agents' actions thus might be drastically different from the actions their current policies would choose. This could cause the agents to converge to sub-optimal actions that appear to be a better choice when considering the effect of potentially arbitrary actions from the other collaborating agents.

In FACMAC, we use a new *centralised* gradient estimator that optimises over the entire joint action space, rather than optimising over each agent's actions separately as in both (3) and (6), to achieve better coordination among agents. In addition, to overcome relative overgeneralisation, when calculating the policy gradient we sample all actions from all agents' current policies when evaluating $Q_{tot}^{\boldsymbol{\mu}}$. Our centralised policy gradient can thus be estimated as

$$\nabla_\theta J(\boldsymbol{\mu}) = \mathbb{E}_\mathcal{D}\Big[\nabla_\theta \boldsymbol{\mu} \nabla_{\boldsymbol{\mu}} Q_{tot}^{\boldsymbol{\mu}}(\boldsymbol{\tau}, \mu_1(\tau_1), \ldots, \mu_n(\tau_n), s)\Big], \tag{7}$$

where $\boldsymbol{\mu} = \{\mu_1(\tau_1; \theta_1), \ldots, \mu_n(\tau_n; \theta_n)\}$ is the set of all agents' current policies and all agents share the same actor network parameterised by $\theta$. However, it is not a requirement of our method for all agents to share parameters in this manner.

If the critic factorisation is linear, as in FACMAC-vdn, then the centralised gradient is equivalent to the per-agent gradients that optimise over each agent's actions separately. This is explored in more detail by DOP [46], which restricts the factored critic to be linear to exploit this equivalence. A major benefit of our method then, is that it does not place any such restrictions on the critic. As remarked by Lyu et al. [24], merely using a centralised critic with per-agent gradients does not necessarily lead to better coordination between agents due to the two problems outlined above. Even with a factored critic, methods that use stochastic policy gradients can still suffer from the problems caused by the per-agent gradients if a fully factored stochastic policy is used. Our centralised gradient estimator, which uses deterministic policies and optimises over the entire joint action space, is required in order to fully take advantage of a centralised critic.

Figure 1 illustrates the overall FACMAC architecture. For each agent $a$, there is one policy network that decides which individual action (discrete or continuous) to take. There is also one critic network for each agent $a$ that estimates the individual agent utilities $Q_a$, which are then combined into the joint action-value function $Q_{tot}$ via a non-linear monotonic mixing function as in QMIX. $Q_{tot}$ is then used by our centralised gradient estimator to help the actor update its policy parameters.

To show the benefits of our new centralised gradient estimator, we compare MADDPG with the centralised policy gradient (CPG) against the original MADDPG on a simple continuous cooperative matrix game with two agents. Figure 10 in Appendix D.1 illustrates this matrix game. There is a

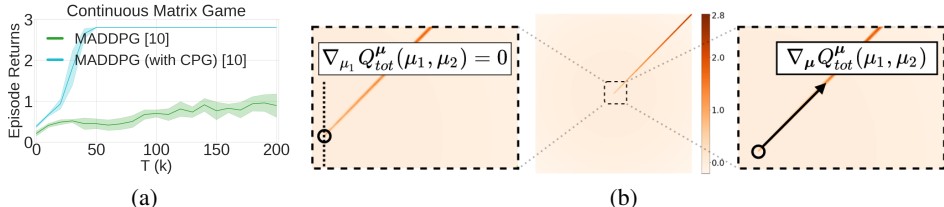

(a)  (b)

Figure 2: (a) Mean test return on Continuous Matrix Game. (b) **Left:** Per-agent policy gradient at the origin. For agent 1 (similarly for agent 2) it is 0 since the gradient term assumes the other agent's action to be fixed and thus it only considers the relative improvements along the dotted line (agent 1's own action space). **Right:** Our centralised policy gradient correctly determines the gradient for improving the joint action.

narrow path (shown in red) starting from the origin $(0, 0)$ to $(1, 1)$, in which the reward gradually increases. Everywhere else there is a small punishment moving away from the origin, increasing in magnitude further from the origin. Experimental results are shown in Figure 2(a). MADDPG quickly gets stuck in the local optimum within $200k$ timesteps, while MADDPG (with CPG) robustly converges to the optimal policy. Figure 2(b) visualises the differences between the per-agent and centralised policy gradients, demonstrating that the per-agent policy gradient can be wrong in our continuous matrix game and our centralised policy gradient is necessary to take advantage of the centralised critic. In Section 5, we further demonstrate the benefits of our centralised gradient estimator in more complex tasks.

### 3.3  Discrete Policy Learning

As FACMAC requires differentiable policies and the sampling process of discrete actions from a categorical distribution is not differentiable, we use the Gumbel-Softmax estimator [14] to enable efficient learning of FACMAC on cooperative tasks with discrete actions. The Gumbel-Softmax estimator is a continuous distribution that approximates discrete samples from a categorical distribution to produce differentiable samples. Moreover, we use the Straight-Through Gumbel-Softmax Estimator [14] to ensure the action dynamics during training and evaluation are the same. Specifically, during training, we sample discrete actions $u_a$ from the original categorical distribution in the forward pass, but use the continuous Gumbel-Softmax sample $x_a$ in the backward pass to approximate the gradients: $\nabla_{\theta_a} u_a \approx \nabla_{\theta_a} x_a$. We can then update the agent's policy using our centralised policy gradient: $\nabla_\theta J(\boldsymbol{x}) \approx \mathbb{E}_{\mathcal{D}}\Big[\nabla_\theta \boldsymbol{x} \nabla_{\boldsymbol{x}} Q_{tot}^{\boldsymbol{x}}(\boldsymbol{\tau}, x_1, \ldots, x_n, s)\Big]$, where $\boldsymbol{x} = \{x_1, \ldots, x_n\}$ is the set of continuous samples that approximates the discrete agent actions. The softmax temperature hyperparameter $\tau$ is set to be 1 in our experiments.

## 4  Multi-Agent MuJoCo

The evaluation of continuous MARL algorithms has recently been largely limited to the simple multi-agent particle environments [23]. We believe the lack of diverse continuous benchmarks is one factor limiting progress in continuous MARL. To demonstrate FACMAC's scalability to more complex continuous domains and to stimulate more progress in continuous MARL, we develop *Multi-Agent MuJoCo* (MAMuJoCo), a novel benchmark for continuous cooperative multi-agent robotic control. Starting from the popular fully observable single-agent robotic MuJoCo [42] control suite included with OpenAI Gym [5], we create a wide variety of novel scenarios in which multiple agents within a single robot have to solve a task cooperatively.

This design offers important benefits. It facilitates comparisons to existing literature on both the fully observable single-agent domain [29], as well as settings with low-bandwidth communication [45]. More importantly, it allows for the study of novel MARL algorithms for decentralised coordination in isolation (scenarios with multiple robots may add confounding factors such as spatial exploration), which is currently a gap in the research literature. MAMuJoCo also includes scenarios with a larger and more flexible number of agents, which takes inspiration from modular robotics [49, 19]. Compared to traditional robots, modular robots are more versatile, configurable, and scalable. We thus

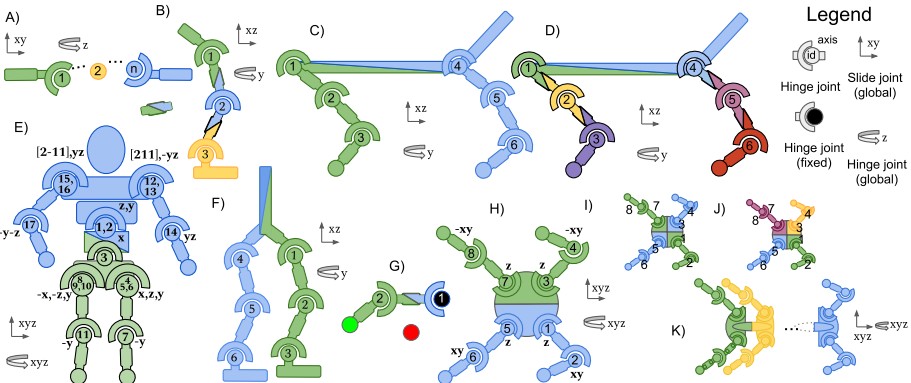

Figure 3: **Agent partitionings for MAMuJoCo environments:** A) Manyagent Swimmer, B) 3-Agent Hopper [3x1], C) 2-Agent HalfCheetah [2x3], D) 6-Agent HalfCheetah [6x1], E) 2-Agent Humanoid and 2-Agent HumanoidStandup (each [1x9,1x8]), F) 2-Agent Walker [2X3], G) 2-Agent Reacher [2x1], H) 2-Agent Ant [2x4], I) 2-Agent Ant Diag [2x4], J) 4-Agent Ant [4x2], and K) Manyagent Ant. Colours indicate agent partitionings. Each joint corresponds to a single controllable motor. Split partitions indicate shared body segments. Square brackets indicate [(number of agents) x (joints per agent)]. Joint IDs are in order of definition in the corresponding OpenAI Gym XML asset files [5]. Global joints indicate degrees of freedom of the center of mass of the composite robotic agent.

develop two scenarios named ManyAgent Swimmer and ManyAgent Ant, in which one can configure an arbitrarily large number of agents (within the memory limits), each controlling a consecutive segment of arbitrary length.

Single-robot multi-agent tasks in MAMuJoCo arise by first representing a given single robotic agent as a *body graph*, where vertices (joints) are connected by adjacent edges (body segments), as shown in Figure 3. We then partition the body graph into disjoint sub-graphs, one for each agent, each of which contains one or more joints that can be controlled. Note that in ManyAgent Swimmer (see Figure 3A) and ManyAgent Ant (see Figure 3K), the number of agents are not limited by the given single robotic agent. See Appendix B for more details about MAMuJoCo.

## 5  Experimental Results

In this section we present our experimental results on our cooperative variants of the continuous *simple tag* environment introduced by Lowe et al. [23] (we refer to this environment as Continuous Predator-Prey), our novel continuous benchmark MAMuJoCo, and the challenging SMAC[6] [35] benchmark with discrete action spaces. In discrete cooperative tasks, we compare with state-of-the-art multi-agent actor-critic algorithms MADDPG [23], COMA [9], CentralV [9], DOP [46], VDAC-mix [40], and value-based methods QMIX [31] and QPLEX [44]. In continuous cooperative tasks, we compare with MADDPG [23] and independent DDPG (IDDPG), as well as COVDN and COMIX, two novel baselines described below. We also explore different forms of critic factorisation to better understand the advantages of factoring a centralised critic. More details about the environments, experimental setup, and training details are included in Appendix D and E. Code is available at https://github.com/oxwhirl/facmac.

**COVDN and COMIX.** We find that not many multi-agent value-based methods work off the shelf with continuous actions. To compare FACMAC against value-based approaches in continuous cooperative tasks, we use existing continuous $Q$-learning approaches in single-agent settings to extend VDN and QMIX to continuous action spaces. Specifically, we introduce COVDN and COMIX, which use VDN-style and QMIX-style factorisation respectively and both perform approximate greedy action selection using the cross-entropy method (CEM) [7]. CEM is a sampling-based derivative-free

---

[6]We utilise SC2.4.10., which is used by the latest PyMARL [35] framework. The original results reported in Samvelyan et al. [35] and Rashid et al. [33] use SC2.4.6. Performance is **not** always comparable across versions.

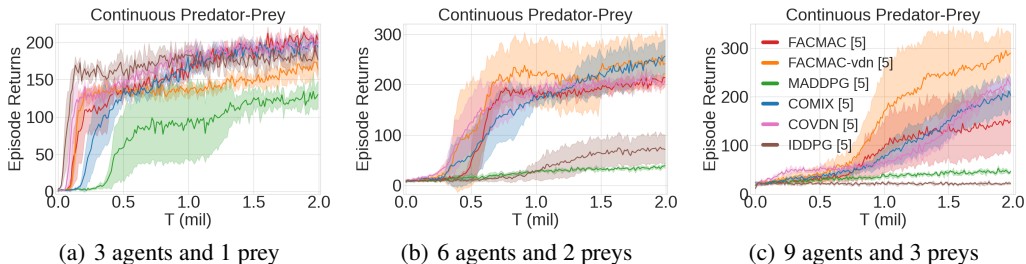

|                    |                     |                     |
|:------------------:|:-------------------:|:-------------------:|
| (a) 3 agents and 1 prey | (b) 6 agents and 2 preys | (c) 9 agents and 3 preys |

Figure 4: Mean episode return on Continuous Predator-Prey with different number of agents and preys. The mean across 5 seeds is plotted and the 95% confidence interval is shown shaded. The numbers in square brackets in the figure legend represent the number of random seeds used to run each method (similarly for all other figures).

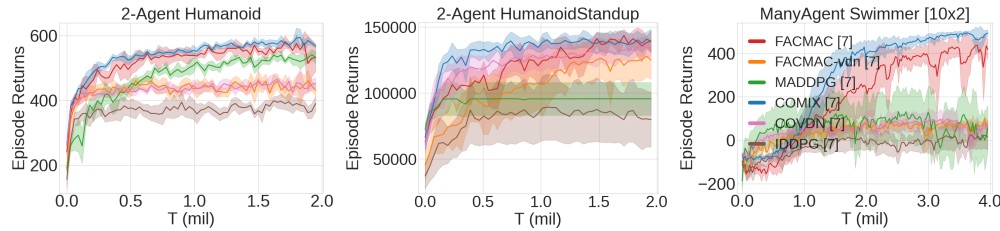

Figure 5: Mean episode return on different MAMuJoCo tasks. In ManyAgent Swimmer, we configure the number of agents to be 10, each controlling a consecutive segment of length 2. The mean across 7 seeds is plotted and the 95% confidence interval is shown shaded.

heuristic search method that has been successfully used to find approximate maxima of nonconvex $Q$-networks in single-agent robotic control tasks [15]. The centralised but factored $Q_{tot}$ allows us to use CEM to sample actions for each agent independently and to use the per-agent utility $Q_a$ to guide the selection of maximal actions. We do not consider COVDN and COMIX significant algorithmic contributions but instead merely baseline algorithms. See Appendix C for more details about them.

**FACMAC outperforms MADDPG and other baselines in both discrete and continuous action tasks.** Figure 4 and 5 illustrate the mean episode return attained by different methods on Continuous Predator-Prey with varying number of agents and different MAMuJoCo tasks, respectively. We can see that FACMAC significantly outperforms MADDPG on all these continuous cooperative tasks, both in terms of absolute performance and learning speed. On discrete SMAC tasks, Figure 6 shows that FACMAC performs significantly better than MADDPG on 4 out of 6 maps we tested, and achieves similar performance to MADDPG on the other 2 maps. Additionally, on all 6 SMAC maps, FACMAC significantly outperforms all multi-agent actor-critic baselines (COMA, CentralV, DOP, and VDAC-mix), while DOP is recently claimed to be the first multi-agent actor-critic method that outperforms state-of-the-art valued-based methods on SMAC. FACMAC is also competitive with state-of-the-art value-based methods (QMIX and QPLEX), with significantly better performance on *MMM*, *bane_vs_bane*, *MMM2*, and *27m_vs_30m*. These results demonstrate the benefits of our method for improving performance in challenging cooperative tasks with discrete and continuous action spaces.

In Continuous Predator-Prey, FACMAC-vdn scales better than FACMAC when the number of agents increases. However, on MAMuJoCo, FACMAC-vdn performs drastically worse than FACMAC in 2-Agent Humanoid and ManyAgent Swimmer (with 10 agents), demonstrating the necessity of the non-linear mixing of agent utilities and conditioning on the central state information in order to achieve competitive performance in such tasks. Furthermore, on SMAC, Figure 13 in Appendix F shows that FACMAC is noticeably more stable than FACMAC-vdn and FACMAC-vdn-s across different maps, and achieves significantly better performance on the *super hard* map *MMM2*.

Interestingly, we find that FACMAC performs similarly to COMIX on both Continuous Predator-Prey and MAMuJoCo tasks. As FACMAC and COMIX use the same value factorisation as in QMIX and are both off-policy, this suggests that, in these continuous cooperative tasks, factorisation of the

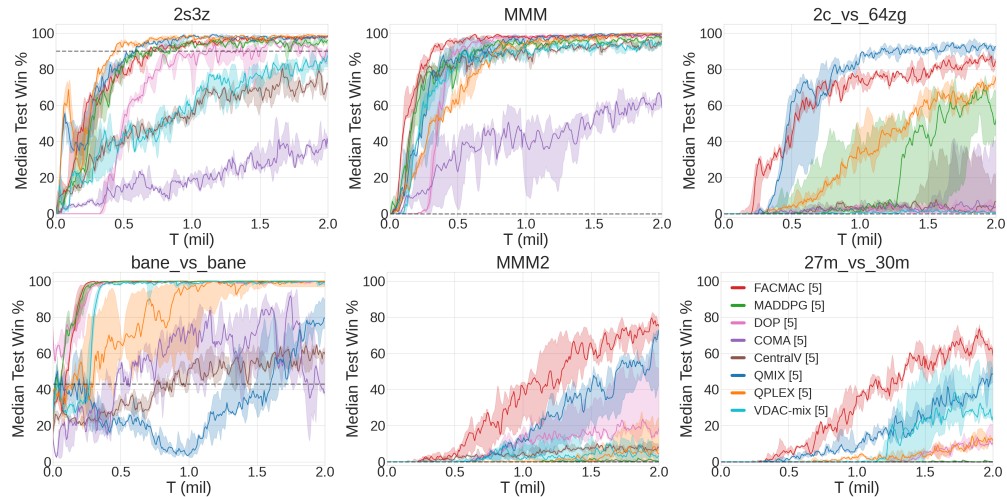

Figure 6: Median test win % on six different SMAC maps, including *2s3z* (easy), *MMM* (easy), *2c_vs_64zg* (hard), *bane_vs_bane* (hard), *MMM2* (super hard), and *27m_vs_30m* (super hard). The median across 5 seeds is plotted and the $25 - 75\%$ percentiles is shown shaded. The performance of the heuristic-based algorithm is shown as a dashed line. We report the median instead of the mean as recommended by Samvelyan et al. [35] in order to avoid the effect of any outliers.

joint $Q$-value function plays a greater role in performance than the underlying algorithmic choices. On SMAC, however, FACMAC performs significantly better than QMIX on *MMM*, *bane_vs_bane*, *MMM2*, and *27m_vs_30m*. For instance, on *bane_vs_bane*, a task with 24 agents, while QMIX struggles to find the optimal policy with 2 million timesteps, FACMAC, with exactly the same value factorisation, can quickly recover the optimal policy and achieve $100\%$ test win rate. This shows the convergence advantages of policy gradient methods in this type of multi-agent settings [43].

**FACMAC scales better as the number of agents (and/or actions) and the complexity of the task increases.** As shown in Figure 4(b) and 4(c), MADDPG performs poorly if we increase the number of agents in Continuous Predator-Prey, while both FACMAC and FACMAC-vdn achieve significantly better performance. The monolithic critic in MADDPG simply concatenates all agents' observations into a single input vector, which can be quite large when there are many agents and/or entities and make it more difficult to learn a good critic. Factoring the critic enables scalable critic learning, by combining individual agent utilities that condition on much smaller observations into a joint action-value function. On MAMuJoCo (shown in Figure 5), similarly, the largest performance gap between FACMAC and MADDPG can be seen on ManyAgent Swimmer (with 10 agents), a task with the largest number of agents among three MAMuJoCo tasks tested.

On SMAC (shown in Figure 6), the largest performance gap between FACMAC and MADDPG can be seen on the challenging *MMM2* and *27m_vs_30m* with a large number of agents, which are classified as 2 *super hard* SMAC maps due to current methods' poor performance [35]. We can see that FACMAC is able to learn to consistently defeat the enemy, whereas MADDPG fails to learn anything useful in both tasks. The second largest performance gap between FACMAC and MADDPG can be seen on the hard map *2c_vs_64zg*, where MADDPG not only performs significantly worse but also exhibits significantly more variance than FACMAC across seeds. While there are only 2 agents in this scenario, the number of actions each agent can choose is the largest among all 6 maps tested as there are 64 enemies. These results further demonstrate that FACMAC scales better when the number of agents (and/or actions) and the complexity of the tasks increases.

**Factoring the critic can better take advantage of our centralised gradient estimator to optimise the agent policies when the number of agents and/or actions is large.** We conduct ablation experiments to investigate the influence of factoring the critic and using the centralised gradient estimator in our method. FACMAC (without CPG) is our method without the centralised policy gradient. It uses a naive adaptation of the deterministic policy gradient used in MADDPG (shown in (6)). Thus, the only difference between FACMAC (without CPG) and MADDPG is that the previous

one learns a non-linearly factored critic while the latter one learns a monolithic critic. We also evaluate MADDPG with our centralised policy gradient and refer to it as MADDPG (with CPG).

Figure 7 shows the ablation results on SMAC and MAMuJoCo. We can see that FACMAC (without CPG) performs significantly better than MADDPG on both SMAC maps tested, demonstrating the advantages of factoring the critic in challenging coordination problems. With the centralised policy gradient, MADDPG (with CPG) performs significantly better than MADDPG on *2c_vs_64zg*. However, on the harder map *MMM2*, MADDPG with both policy gradients fail to learn anything useful. By contrast, FACMAC significantly outperforms FACMAC (without CPG) on *MMM2*, and has lower variance across seeds on *2c_vs_64zg*. Furthermore, on ManyAgent Swimmer with 2 agents, our centralised gradient esti-

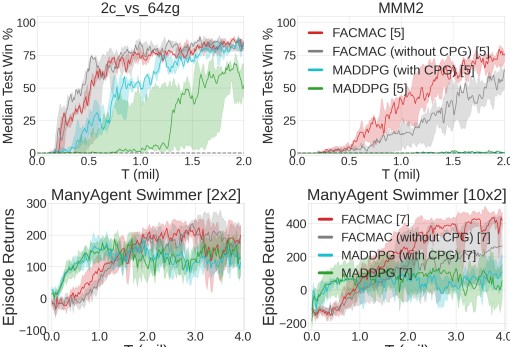

Figure 7: Ablations for different FACMAC components on SMAC and MAMuJoCo tasks.

mator does not affect the performance of both MADDPG and FACMAC. However, on the same task with 10 agents, using the centralised gradient estimator significantly improves the learning performance when learning a factored critic. These results demonstrate that factoring the critic can better take advantage of our centralised gradient estimator to optimise the agent policies when the number of agents and/or actions is large.

**Nonmonotonically factored critics can solve tasks that cannot be solved with monolithic or monotonically factored critics.** In our multi-agent actor-critic framework, there are no inherent constraints on factoring the critic, we thus also employ a nonmonotonic factorisation and refer to it as FACMAC-nonmonotonic (as discussed in Section 3.1). As shown in Figure 8 (left), on our continuous matrix game (as discussed in Section 3.2), FACMAC-nonmonotonic can robustly learn the optimal policy, while both

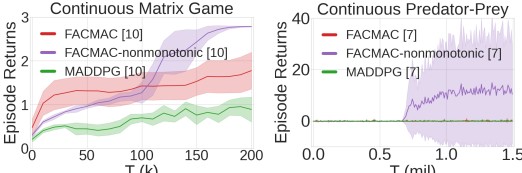

Figure 8: Mean episode return on (**Left**) Continuous Matrix Game and (**Right**) a variant of our Continuous Predator-Prey task (with 3 agents and 1 prey) with *nonmonotonic* value functions.

FACMAC and MADDPG converge to some sub-optimal policy within $200k$ timesteps. On a variant of Continuous Predator-Prey task (see Appendix D.2 for more details about this task) with *nonmonotonic* value functions (i.e., an agent's ordering over its own actions depends on other agents' actions [25]), Figure 8 (right) shows that both FACMAC and MADDPG fail to learn anything useful, while FACMAC-nonmonotonic successfully learns to capture the prey. These results demonstrate that nonmonotonically factored critics can solve tasks that cannot be solved with monolithic or monotonically factored critics.

It is important to note that the relative performance of FACMAC and FACMAC-nonmonotonic is task dependent. On the original Continuous Predator-Prey task (with 3 agents and 1 prey), FACMAC-nonmonotonic yields similar performance to FACMAC (see Figure 12 in Appendix F). On SMAC (see Figure 13 in Appendix F), FACMAC-nonmonotonic performs similarly to FACMAC on easy maps, but exhibits significantly worse performance on harder maps. This shows that, in this type of tasks, using an unconstrained factored critic could lead to an increase in learning difficulty. We thus expect FACMAC-nonmonotonic to be more useful in tasks with nonmonotonic value functions.

## 6   Conclusion

This paper presented FACMAC, a multi-agent actor-critic method that learns decentralised policies with a centralised but factored critic, working for both discrete and continuous cooperative tasks. We showed the advantages of both factoring the critic and using the new centralised gradient estimator in our approach. We also introduced a novel benchmark suite MAMuJoCo to demonstrate FACMAC's scalability to more complex continuous tasks. Our results on three different domains demonstrated FACMAC's superior performance over existing MARL algorithms. Future work will explore more forms of nonmonotonic factorisation to tackle tasks with nonmonotonic value functions.

## Acknowledgements

We would like to thank Dr Martin Strohmeier and the members of the Whiteson Research Lab for their helpful feedback. We would also like to thank the anonymous reviewers for their constructive comments during the reviewing process. This project has received funding from the European Research Council (ERC), under the European Union's Horizon 2020 research and innovation programme (grant agreement number 637713). It was also supported by an EPSRC grant (EP/M508111/1, EP/N509711/1). Christian Schroeder de Witt is generously funded by Cyber Defence Campus, Science and Technology, Armasuisse, Switzerland. The experiments were made possible by a generous equipment grant from NVIDIA.

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
