# A   Related Work

Value function factorisation [17] has been widely employed in value-based MARL algorithms. VDN [41] and QMIX [31] factor the joint action-value function into per-agent utilities that are combined via a simple summation or a monotonic mixing function respectively, to ensure consistency between the $\arg\max$ of the centralised joint-action value function and the $\arg\max$ of the decentralised polices. This monotonicity constraint, however, prevents them from representing joint action-value functions that are characterised as *nonmonotonic* [25], i.e., an agent's ordering over its own actions depends on other agents' actions. A large number of recent works [37, 48, 32, 44, 38] thus focus on developing new value-based MARL algorithms that address this representational limitation, in order to learn a richer class of action-value functions.

QTRAN [37] learns an unrestricted joint action-value function and aims to solve a constrained optimisation problem in order to decentralise it, but has been shown to scale poorly to more complex tasks such as SMAC [25]. QPLEX [44] takes advantage of the dueling network architecture to factor the joint action-value function in a manner that does not restrict the representational capacity, whilst also remaining easily decentralisable, but can still fail to solve simple tasks with nonmonotonic value functions [32]. Weighted QMIX [32] introduces a weighting scheme to place more importance on the better joint actions to learn a richer class of joint action-value functions. QTRAN++ [38] addresses the gap between the empirical performance and theoretical guarantees of QTRAN. Our multi-agent actor-critic framework with decentralised actors and a centralised but factored critic, by contrast, provides a more direct and simpler way of coping with nonmonotonic tasks as one can simply factor the centralised critic in any manner without constraints. Additionally, our framework can be readily applied to tasks with continuous action spaces, whereas these value-based algorithms require additional algorithmic changes.

Most state-of-the-art multi-agent actor-critic methods [23, 9, 13, 8] learn a centralised and monolithic critic conditioning on the global state and the joint action to stabilise learning. Even though the joint action-value function they can represent is not restricted, in practice they significantly underperform value-based methods like QMIX on the challenging SMAC benchmark [32, 33]. In contrast, FACMAC utilises a centralised but factored critic to allow it to scale to the more complex tasks in SMAC, and follows the centralised policy gradient instead of per-agent policy gradients.

Lyu et al. [24] recently provide some interesting insights about the pros and cons of centralised and decentralised critics for on-policy actor-critic algorithms that use stochastic policy gradients. One important issue that they highlight is that merely utilising a centralised critic does not necessarily lead to the learning of more coordinated behaviours than using decentralised critics. This is because the use of a per-agent policy gradient can lead to the agents getting stuck in sub-optimal solutions in which no agents wishes to change their policy, as discussed in Section 3.2. Su et al. [40] recently propose the value-decomposition actor-critic (VDAC) framework that factors the centralised critic monotonically (i.e., factors the joint value function as a monotonic combination of local value functions) to learn more efficiently. However, even with a factored critic, VDAC still suffers from the problems caused by the per-agent policy gradient due to the use of a fully factored stochastic policy. Our centralised policy gradient resolves this issue by using deterministic policies and taking full advantage of the centralised training paradigm to optimise over the joint-action policy, which allows us to fully reap the benefits of a centralised critic. Furthermore, VDAC requires on-policy learning, which can be sample-efficient. In contrast, FACMAC is off-policy, we thus benefit immensely from utilising a centralised critic over a decentralised one since we avoid the issues of non-stationarity when training on older data.

Zhou et al. [50] propose to use a single centralised critic for MADDPG, whose weights are generated by hypernetworks that condition on the state, similarly to QMIX's mixing network without the monotonicity constraints. FACMAC also uses a single centralised critic, but factorises it similarly to QMIX (not just using the mixing network) which allows for more efficient learning on more complex tasks. Of existing work, the deterministic decomposed policy gradients (DOP) algorithm proposed by Wang et al. [46] is perhaps most similar to our own approach. Deterministic DOP is off-policy and factors the centralised critic as a weighted linear sum of individual agent utilities and a state bias. It is limited to only considering linearly factored critics, which have limited representational capacity, whilst we are free to choose any method of factorisation in FACMAC to allow for the learning of a richer class of action-value functions. While they claim to be the first to introduce the idea of value function factorisation into the multi-agent actor-critic framework, it is actually first explored

by Bescuca [4], where a monotonically factored critic is learned for COMA [9]. However, their performance improvement on SMAC is limited since COMA requires on-policy learning and it is not straightforward to extend COMA to continuous action spaces. Furthermore, both works (and VDAC) only consider monotonically factored critics, whilst we employ a nonmonotonic factorisation and demonstrate its benefits. We also investigate the benefits of learning a centralised but factored critic more thoroughly, providing a better understanding about the type of tasks that can benefit more from a factored critic. Furthermore, both deterministic DOP and LICA [50] use a naive adaptation of the deterministic policy gradient used by MADDPG and suffer from the same problems as discussed in Section 3.2, while our centralised policy gradients allow for better coordination across agents in certain tasks.

## B Multi-Agent MuJoCo

While several MARL benchmarks with continuous action spaces have been released, few are simultaneously diverse, fully cooperative, decentralisable, and admit partial observability. The multi-agent particle suite [23] features a few decentralisable tasks in a fully observable planar point mass toy environment. Presumably due to its focus on real-world robotic control, RoboCup soccer simulation [16, 39, 34] does not currently feature an easily configurable software interface for MARL, nor suitable AI-controlled benchmark opponents. Liu et al. [22] introduce MuJoCo soccer environment, a multi-agent soccer environment with continuous simulated physics that cannot be used in a purely cooperative setting and does not admit partial observability.

To demonstrate FACMAC's scalability to more complex continuous domains and to stimulate more progress in continuous MARL, we develop *Multi-Agent MuJoCo* (MAMuJoCo), a novel benchmark for continuous cooperative multi-agent robotic control. Starting from the popular fully observable single-agent robotic MuJoCo [42] control suite included with OpenAI Gym [5], we create a wide variety of novel scenarios in which multiple agents within a single robot have to solve a task cooperatively. Single-robot multi-agent tasks in MAMuJoCo arise by first representing a given single robotic agent as a *body graph*, where vertices (joints) are connected by adjacent edges (body segments), as shown in Figure 3. We then partition the body graph into disjoint sub-graphs, one for each agent, each of which contains one or more joints that can be controlled. Note that in ManyAgent Swimmer (see Figure 3A) and ManyAgent Ant (see Figure 3K), the number of agents are not limited by the given single robotic agent.

Multiple agents are introduced within a single robot as partial observability arises through latency, bandwidth, and noisy sensors in a single robot. Even if communication is free and instant when it works, we want policies that keep working even when communication channels within the robot malfunction. Without access to the exact full state, local decision rules become more important and introducing autonomous agents at individual decision points (e.g., each physical component of the robot) is reasonable and beneficial. This also makes it more robust to single-point failures (e.g., broken sensors) and more adaptive and flexible as new independent decision points (thus agents) may be added easily. Similar physical decompositions can be found in early behavior-based robotic research [6, 3].

Each agent's action space in MAMuJoCo is given by the joint action space over all motors controllable by that agent. For example, the agent corresponding to the green partition in 2-Agent HalfCheetah (Figure 3C) consists of three joints (joint ids 1, 2, and 3) and four adjacent body segments. Each joint has an action space $[-1, 1]$, so the action space for each agent is a 3-dimensional vector with each entry in $[-1, 1]$.

For each agent $a$, observations are constructed in a two-stage process. First, we infer which body segments and joints are observable by agent $a$. Each agent can always observe all joints within its own sub-graph. A configurable parameter $k \geq 0$ determines the maximum graph distance to the agent's subgraph at which joints are observable (see Figure 9 for an example). Body segments directly attached to observable joints are themselves observable. The agent observation is then given by a fixed order concatenation of the representation vector of each observable graph element. Depending on the environment and configuration, representation vectors may include attributes such as position, velocity, and external body forces. In addition to joint and body-segment specific observation categories, agents can also be configured to observe the position and/or velocity attributes of the robot's central torso.

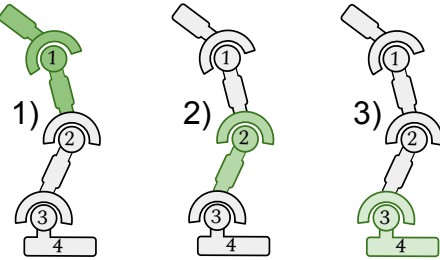

Figure 9: **Observations by distance for 3-Agent Hopper (as seen from agent 1).** Each corresponds to joints and body parts observable at 1) zero graph distance from agent 1, 2) one unit graph distance from agent 1, and 3) two unit graph distances from agent 1.

Restricting both the observation distance $k$, as well as limiting the set of observable element categories imposes partial observability. However, task goals remain unchanged from the single-agent variants, except that the goals must be reached collaboratively by multiple agents: we simply repurpose the original single-agent reward signal as a team reward signal. Default team reward is summarised in Table 1.

| Task | Goal | Special observations | Reward function |
|---|---|---|---|
| 2-Agent Swimmer | Maximise +ve $x$-speed. | All agents can observe velocities of the central torso. | $\frac{\Delta x}{\Delta t} + 0.0001\alpha$ |
| 2-Agent Reacher | Fingertip (green) needs to reach target at random location (red). | Target is only visible to green agent. | $-\,\|\text{distance from fingertip to target}\|_2^2$ $+\alpha$ |
| 2-Agent Ant | Maximise +ve $x$-speed. | All agents can observe velocities of the central torso. | $\frac{\Delta x}{\Delta t} + 5 \cdot 10^{-4} \,\|\text{external contact forces}\|_2^2$ $+0.5\alpha + 1$ |
| 2-Agent Ant Diag | Maximise +ve $x$-speed. | All agents can observe velocities of the central torso. | $\frac{\Delta x}{\Delta t} + 5 \cdot 10^{-4} \,\|\text{external contact forces}\|_2^2$ $+0.5\alpha + 1$ |
| 2-Agent HalfCheetah | Maximise +ve $x$-speed. | - | $\frac{\Delta x}{\Delta t} + 0.1\alpha$ |
| 2-Agent Humanoid | Maximise +ve $x$-speed. | - | $0.25\frac{\Delta x}{\Delta t} + \min(10,$ $5 \cdot 10^{-6} \,\|\text{external contact forces}\|_2^2)$ |
| 2-Agent HumanoidStandup | Maximise +ve $x$-speed. | - | $\frac{y}{\Delta t} + \min(10,$ $5 \cdot 10^{-6} \,\|\text{external contact forces}\|_2^2)$ |
| 3-Agent Hopper | Maximise +ve $x$-speed. | - | $\frac{\Delta x}{\Delta t} + 0.001\alpha + 1.0$ |
| 4-Agent Ant | Maximise +ve $x$-speed. | All agents can observe velocities of the central torso. | $\frac{\Delta x}{\Delta t} + 5 \cdot 10^{-4} \,\|\text{external contact forces}\|_2^2$ $+0.5\alpha + 1$ |
| 6-Agent HalfCheetah | Maximise +ve $x$-speed. | - | $0.25\frac{\Delta x}{\Delta t} + \min(10,$ $5 \cdot 10^{-6} \,\|\text{external contact forces}\|_2^2)$ |
| ManyAgent Swimmer | Maximise +ve $x$-speed. | All agents can observe velocities of the central torso. | $\frac{\Delta x}{\Delta t} + 0.0001\alpha$ |
| ManyAgent Ant | Maximise +ve $x$-speed. | All agents can observe velocities of the central torso. | $\frac{\Delta x}{\Delta t} + 5 \cdot 10^{-4} \,\|\text{external contact forces}\|_2^2$ $+0.5\alpha + 1$ |

Table 1: Overview of tasks contained in MAMuJoCo. We define $\alpha$ as an action regularisation term $-\|\mathbf{u}\|_2^2$.

The most similar existing environments, though not as diverse as MAMuJoCo, are the decomposed MuJoCo environments Centipede and Snakes [45]. The latter is similar to MAMuJoCo's 2-Agent Swimmer. Ackermann et al. [1] evaluate on one environment similar to a configuration of 2-Agent Ant, but, similarly to Gupta et al. [11], do not consider tasks across different numbers of agents and MuJoCo scenarios.

### B.1 Social Impact

MAMuJoCo is a simulated environment that mimics certain fundamental aspects of robotic actuator chains. Such actuator chains are commonly found in industrial robotics, perhaps most prominently in robot arms on assembly lines, but also a variety of other civil use cases, including remote space exploration or bomb disposal. However, in order to be applicable to such use cases, policies trained in MAMuJoCo first need to undergo a transfer process to the real world, for example by using Sim2Real techniques. MAMuJoCo can not only help assess algorithmic advances, but also contribute toward understanding and mitigating risks inherent to civil autonomous robotics applications. For example, MAMuJoCo could facilitate the development of safe reinforcement learning algorithms that keep

actuator parameters of real-world robotic arms within safe ranges, thus avoiding manufacturing errors or injury.

As most developments in robotic control, algorithmic progress based on our benchmark environment, or the novel algorithms introduced in this paper, might ultimately find application in autonomous warfare. MAMuJoCo has been developed with fundamental algorithmic development for civil purposes in mind and does not provide any features that would make it particularly suitable to non-civil use. Any robotic agents trained with reinforcement learning algorithms on our environment, including our novel methods, should be certified with respect to fairness and security before real-world deployment. Like any artificial intelligence system, our proposed framework has the potential to greatly improve human productivity. However, it may also reduce the need for human workers, resulting in job losses.

## C   COVDN and COMIX

$Q$-learning has shown considerable success in multi-agent settings with discrete action spaces [41, 31, 37, 32, 44]. However, performing greedy action selection in $Q$-learning requires evaluating $\arg\max_{\mathbf{u}} Q_{tot}(\boldsymbol{\tau}, \mathbf{u}, s)$, where $Q_{tot}$ is the joint action-value function. In discrete action spaces, this operation can be performed efficiently through enumeration (unless the action space is extremely large). In continuous action spaces, however, enumeration is impossible. Hence, existing continuous $Q$-learning approaches in single-agent settings either impose constraints on the form of $Q$-value to make maximisation easy [10, 2], at the expense of estimation bias, or perform only approximate greedy action selection [15]. Neither approach scales easily to the large joint action spaces inherent to multi-agent settings, as 1) the joint action space grows exponentially in the number of agents, and 2) training $Q_{tot}$ required for greedy action selection becomes impractical when there are many agents.

This highlights the importance of learning a centralised but factored $Q_{tot}$. To factor large joint action spaces efficiently in a decentralisable fashion, COVDN represents the joint action-value function $Q_{tot}$ as a sum of the per-agent utilities $Q_a$ as in VDN [41], while COMIX represents $Q_{tot}$ as a non-linear monotonic combination of $Q_a$ as in QMIX [31]. COVDN and COMIX are thus simple variants of VDN and QMIX, respectively, that scale to continuous action spaces. They both perform approximate greedy selection of actions $u_a$ with respect to utility functions $Q_a$ for each agent $a$ using the cross-entropy method (CEM) [7]. CEM is a sampling-based derivative-free heuristic search method that has been successfully used to find approximate maxima of nonconvex $Q$-networks in a number of single-agent robotic control tasks [15]. The centralised but factored $Q_{tot}$ allows us to use CEM to sample actions for each agent independently and to use the individual utility function $Q_a$ to guide the selection of maximal actions.

In both COVDN and COMIX, CEM is used by each agent $a$ to find an action that approximately optimises its local utility function $Q_a$. Specifically, CEM iteratively draws a batch of $N$ random samples from a candidate distribution $\mathcal{D}_k$, e.g., a Gaussian, at each iteration $k$. The best $M < N$ samples (with the highest utility values) are then used to fit a new Gaussian distribution $\mathcal{D}_{k+1}$, and this process repeats $K$ times. We use a CEM hyperparameter configuration similar to Qt-Opt [15], where $N = 64$, $M = 6$, and $K = 2$.[7] Gaussian distributions are initialised with mean $\mu = 0$ and standard deviation $\sigma = 1$. Algorithm 1 outlines the full CEM process used in both COVDN and COMIX. Algorithm 2 outlines the full process for COMIX. Note we do not consider COVDN and COMIX significant algorithmic contributions but instead merely baseline algorithms.

## D   Environment Details

### D.1   Continuous Matrix Game

Figure 10 illustrates the continuous matrix game with two agents. There is a narrow path (shown in red) starting from the origin $(0,0)$ to $(1,1)$, in which the reward gradually increases. Everywhere else there is a small punishment moving away from the origin, increasing in magnitude further from the origin.

---

[7]We empirically find 2 iterations to suffice.

**Algorithm 1** For each agent $a$, we perform $n_c$ CEM iterations. Hyper-parameters $d_i \in \mathbb{N}$ control how many actions are sampled at the $i$th iteration.

---

**for** $a := 1,\ a \leq N$ **do**
    $\boldsymbol{\mu}_a \leftarrow \mathbf{0} \in \mathbb{R}^{|\mathcal{A}_a|}$
    $\boldsymbol{\sigma}_a \leftarrow \mathbf{1} \in \mathbb{R}^{|\mathcal{A}_a|}$
    **for** $i := 1,\ i \leq n_c$ **do**
        **for** $j := 1,\ j \leq d_i$ **do**
            $\mathbf{v}'_{aj} \sim \mathcal{N}(\boldsymbol{\mu}_a, \boldsymbol{\sigma}_a)$
            $\mathbf{v}_{aj} \leftarrow \tanh(\mathbf{v}'_{aj})$
            $q_{aj} \leftarrow Q_a(\tau_a, \mathbf{v}_{aj})$
            $j \leftarrow j + 1$
        **end for**
        **if** $i < n_c$ **then**
            $U \leftarrow \{\mathbf{v}'_{al} \mid q_{al} \in \mathrm{top}k_i(q_{a1}, \ldots, q_{ad_i}), \forall l \in \{1 \ldots N\}\}$
            $\boldsymbol{\mu}_a \leftarrow \mathrm{sample\_mean}(U)$
            $\boldsymbol{\sigma}_a \leftarrow \mathrm{sample\_std}(U)$
        **else**
            $m \leftarrow \arg\max_j q_{aj}$
            $\mathbf{u}_a \leftarrow \mathbf{v}_{am}$
        **end if**
        $i \leftarrow i + 1$
    **end for**
    $a \leftarrow a + 1$
**end for**
**return** $\langle \mathbf{u}_1, \ldots, \mathbf{u}_n \rangle$

---

**Algorithm 2** Algorithmic description of COMIX. The function CEM is defined in Algorithm 1.

---

Initialise ReplayBuffer, $\theta, \theta^-, \phi, \phi^-$
**for** each training episode $e$ **do**
    $s_0, \mathbf{z}_0 \leftarrow \mathrm{EnvInit}()$
    **for** $t := 0$ until $t = T$ step $1$ **do**
        $\mathbf{u}_t \leftarrow \mathrm{CEM}(Q_1, \ldots, Q_N, \tau_t^1, \ldots, \tau_t^N)$
        $\langle s_{t+1}, \mathbf{z}_{t+1}, r_t \rangle \leftarrow \mathrm{EnvStep}(\mathbf{u}_t)$
        ReplayBuffer $\leftarrow \langle s_t, \mathbf{u}_t, \mathbf{z}_t, r_t, s_{t+1}, \boldsymbol{z}_{t+1} \rangle$
    **end for**
    $\{\langle s_i, \boldsymbol{u}_i, \boldsymbol{z}_i, r_i, s'_i, \boldsymbol{z}'_i \rangle\}_{i=1}^b \sim \mathrm{ReplayBuffer}$
    $y_i \leftarrow r_i + \gamma \max_{\boldsymbol{u}'_i} Q_{\mathrm{tot}}(s'_i, \boldsymbol{z}'_i, \boldsymbol{u}'_i; \boldsymbol{\theta}^-, \phi^-), \ \forall i$
    $\mathcal{L} \leftarrow \sum_{i=1}^b \left( y_i - Q_{\mathrm{tot}}(s_i, \boldsymbol{z}_i, \boldsymbol{u}_i; \boldsymbol{\theta}, \phi) \right)^2$
    $\boldsymbol{\theta} \leftarrow \boldsymbol{\theta} - \alpha \nabla_{\boldsymbol{\theta}} \mathcal{L}$
    $\phi \leftarrow \phi - \alpha \nabla_\phi \mathcal{L}$
**end for**

---

## D.2 Continuous Predator-Prey

We consider the mixed *simple tag* environment (Figure 11) introduced by Lowe et al. [23], which is a variant of the classic predator-prey game. Three slower cooperating circular agents (red), each with continuous movement action spaces $u^a \in \mathbb{R}^2$, must catch a faster circular prey (green) on a randomly generated two-dimensional toroidal plane with two large landmarks blocking the way.

To obtain a purely cooperative environment, we replace the prey's policy by a hard-coded heuristic, that, at any time step, moves the prey to the sampled position with the largest distance to the closest predator. If one of the cooperative agents collides with the prey, a team reward of $+10$ is emitted; otherwise, no reward is given. In the original simple tag environment, each agent can observe the relative positions of the other two agents, the relative position and velocity of the prey, and the relative

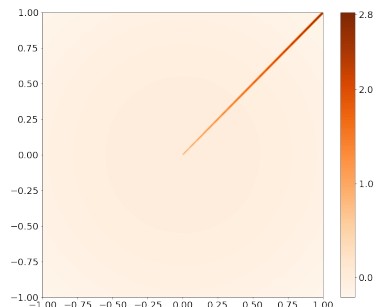

Figure 10: The continuous matrix game.

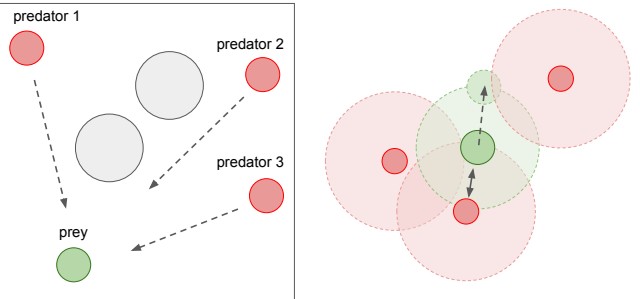

Figure 11: Continuous Predator-Prey. **Left:** Top-down view of toroidal plane, with predators (red), prey (green) and obstacles (grey). **Right:** Illustration of the prey's avoidance heuristic. Observation radii of both agents and prey are indicated.

positions of the landmarks. This means each agent's private observation provides an almost complete representation of the true state of the environment.

To introduce partial observability to the environment, we add an agent *view radius*, which restricts the agents from receiving information about other entities (including all landmarks, the other two agents, and the prey) that are out of range. Specifically, we set the view radius such that the agents can only observe other agents roughly $60\%$ of the time. We refer to this environment as Continuous Predator-Prey.

In addition, we implement a variant of our Continuous Predator-Prey task (with 3 agents and 1 prey), where the reward function is modified to make the task *nonmonotonic*. Specifically, if one agent collides with the prey while at least another one being close enough, a team reward of $+10$ is given. However, if only one agent collides with the prey without any other agent being close enough, a negative team reward of $-1$ is given. Otherwise, no reward is provided.

### D.3 Multi-Agent MuJoCo

All MAMuJoCo environments we tested are configured according to its default configuration, where each agent can observe only positions (not velocities) of its own body parts and at graph distances greater than zero. In ManyAgent Swimmer, we configure the number of agents to be 10, each controlling a consecutive segment of length 2. We thus refer to this environment as ManyAgent Swimmer [10x2]. We set maximum observation distances to $k = 0$ (which means each agent can observe only positions of its own body parts) for all three environments tested, including 2-Agent Humanoid, 2-Agent HumanoidStandup, and ManyAgent Swimer [10x2]. Default team reward is used (see Table 1).

### D.4 SMAC

SMAC consists of a set of complex StarCraft II micromanagement tasks that are carefully designed to study decentralised multi-agent control. The tasks in SMAC involve combat between two armies

of units. The first army is controlled by a group of learned allied agents. The second army consists of enemy units controlled by the built-in heuristic AI. The goal of the allied agents is to defeat the enemy units in battle, to maximise the win rate. The action space consists of a set of discrete actions: `move` in four cardinal directions, `attack` any selected enemy (available if the enemy is within the agent's shooting range), `stop`, and `noop`. Hence the number of actions increases as the number of enemies increases. All experiments on SMAC use the default reward and observation settings of the SMAC benchmark [35].

# E   Experimental Details

We evaluate the performance of each method using the following procedure: for each run of a method, we pause training every fixed number of timesteps (2000 timesteps for Continuous Predator-Prey, 4000 timesteps for MAMuJoCo, and 10000 timesteps for SMAC) and run a fixed number of independent episodes (10 episodes for Continuous Predator-Prey and MAMuJoCo, and 32 episodes for SMAC) with each agent performing action selection greedily in a decentralised fashion. On both Continuous Predator-Prey and MAMuJoCo, the mean value of these episode returns are used to evaluate the performance of the learned policies. On SMAC, we use the median test win rate (i.e., the percentage of the 32 episodes where the agents defeat all enemy units within the permitted time limit) to evaluate the learned policies, as in [35]. All experiments are carried out on NVIDIA GeForce GTX 1080 GPU.

## E.1   Continuous Predator-Prey

In value-based methods COVDN and COMIX, the architecture of the shared agent network is a DRQN with a recurrent layer comprised of a GRU with a $64$-dimensional hidden state, with a fully-connected layer before and after. In actor-critic methods FACMAC, FACMAC-vdn, MADDPG, and IDDPG, the architecture of the shared agent network is also a DRQN with a recurrent layer comprised of a GRU with a $64$-dimensional hidden state, with a fully-connected layer before and after, while the final output layer is a tanh layer, to bound actions. The shared critic network is a MLP with $2$ hidden layers of $64$ units and ReLU non-linearities. All agent networks receive the current local observation and last individual action as input. In MADDPG, the centralised critic receives the global state and the joint action of all agents as input. The global state consists of the joint observations of all agents in Continuous Predator-Prey. In other actor-critic methods, there is a shared critic network that approximates per-agent utilities, which receives each agent's local observation and individual action as input.

During training and testing, we restrict each episode to have a length of $25$ time steps. Training lasts for $2$ million timesteps. To encourage exploration, we use uncorrelated, mean-zero Gaussian noise with $\sigma = 0.1$ during training (for all 2 million timesteps). We set $\gamma = 0.85$ for all experiments. The replay buffer contains the most recent $5000$ transitions. We train on a batch size of $32$ after every timestep. The hyper-parameter *batch_size_run* is set to be 1. For the soft target network updates we use $\tau = 0.001$. All neural networks (actor and critic) are trained using Adam optimiser with a learning rate of $0.01$. To evaluate the learning performance, the training is paused after every 2000 timesteps during which 10 independent test episodes are run with agents performing action selection greedily in a decentralised fashion.

In FACMAC, the architecture of the mixing network consists of a single hidden layer of $64$ units with an ELU non-linearity. The weights of the mixing network are produced by separate hypernetworks. The hypernetworks consist of a feedforward network without any hidden layers. The output of the hypernetwork is passed through an absolute activation function (to achieve non-negativity) and then resized into a matrix of appropriate size. In FACMAC-nonmonotonic, the architecture of the mixing network (and the hypernetworks) is similar to the one used in FACMAC, but without the constraint of monotonicity enforced by using non-negative weights.

## E.2   Multi-Agent MuJoCo

In all value-based methods, the architecture of all agent networks is a MLP with $2$ hidden layers with $400$ and $300$ units respectively, similar to the setting used in OpenAI Spinning Up.[8] All agent

---

[8] `https://spinningup.openai.com/en/latest/`.

networks use ReLU non-linearities for all hidden layers. In all actor-critic methods, the architecture of the shared agent network and critic network is also a MLP with 2 hidden layers with 400 and 300 units respectively, while the final output layer of the actor network is a tanh layer, to bound the actions. In all value-based methods, the agent receives its current local observation as input. In MADDPG, the centralised critic receives the global state and the joint action of all agents as input. The global state consists of the full state information returned by the original OpenAI Gym [5]. In other actor-critic methods, there is a shared critic network that approximates per-agent utilities, which receives each agent's local observation and individual action as input.

During training and testing, we restrict each episode to have a length of 1000 time steps. Training lasts for 2 million or 4 million timesteps. To encourage exploration, we use uncorrelated, mean-zero Gaussian noise with $\sigma = 0.1$ during training. We also use the same trick as in OpenAI Spinning Up to improve exploration at the start of training. For a fixed number of steps at the beginning (we set it to be 10000), the agent takes actions which are sampled from a uniform random distribution over valid actions. After that, it returns to normal Gaussian exploration. We set $\gamma = 0.99$ for all experiments. The replay buffer contains the most recent $10^6$ transitions. We train on a batch size of 100 after every timestep. The hyper-parameter *batch_size_run* is set to be 1. For the soft target network updates we use $\tau = 0.001$. All neural networks (actor and critic) are trained using Adam optimiser with a learning rate of 0.001. To evaluate the learning performance, the training is paused after every 4000 timesteps during which 10 independent test episodes are run with agents performing action selection greedily in a decentralised fashion.

In FACMAC, the architecture of the mixing network consists of a single hidden layer of 64 units with an ELU non-linearity. The weights of the mixing network are produced by separate hypernetworks. The hypernetworks consist of a feedforward network with a single hidden layer of 64 units with a ReLU non-linearity. The output of the hypernetwork is passed through an absolute activation function (to achieve non-negativity) and then resized into a matrix of appropriate size.

### E.3   SMAC

For baseline algorithms DOP [46], COMA [9], CentralV [9], VDAC-mix [40], QMIX [33], and QPLEX [44], we use the the same training setup as provided by their authors where the hyperparameters have been fine-tuned on the SMAC benchmark.

Most of our training hyperparameters for FACMAC and MADDPG [23] follow [33]. In both methods, the architecture of the shared actor network is a DRQN with a recurrent layer comprised of a GRU with a 64-dimensional hidden state, with a fully-connected layer before and after. The shared critic network is a MLP with 2 hidden layers of 64 units and ReLU non-linearities. Exploration is performed during training using a scheme similar to COMA [9]. Action probabilities are produced from the final layer of the actor network, $z$, via a bounded softmax distribution that lower-bounds the probability of any given action by $\epsilon/|U|$: $P(u) = (1 - \epsilon)\text{softmax}_u + \epsilon/|U|$, where $|U|$ is the size of the joint action space. Throughout the training, for exploration we anneal $\epsilon$ linearly from 0.5 to 0.05 over $50k$ timesteps and keep it constant for the rest of the training. On map *MMM2*, we anneal $\epsilon$ linearly from 1.0 to 0.05 over $50k$ timesteps and keep it constant for the rest of the training for FACMAC and MADDPG (as it gives better performance). The replay buffer contains the most recent 5000 episodes. We sample batches of 32 episodes uniformly from the replay buffer and train on fully unrolled episodes. In MADDPG, we use a target network for the actor and critic, respectively. In FACMAC, we use a target network for the actor, critic, and mixing network, respectively. All target networks are periodically updated every 200 training steps. All neural networks are trained using Adam optimiser with learning rate 0.0025 for the actor network and 0.0005 for the critic network. We set $\gamma = 0.99$ for all experiments.

The architecture of the mixing network in FACMAC follows [33]. It consists of a single hidden layer of 32 units with an ELU non-linearity. The weights of the mixing network are produced by separate hypernetworks. The hypernetworks consist of a feedforward network with a single hidden layer of 64 units with a ReLU non-linearity. The output of the hypernetwork is passed through an absolute activation function (to achieve non-negativity) and then resized into a matrix of appropriate size. In FACMAC-nonmonotonic, the architecture of the mixing network (and the hypernetworks) is similar to the one used in FACMAC, but without the constraint of monotonicity enforced by using non-negative weights.

# F    Additional Results on Different Critic Factorisations

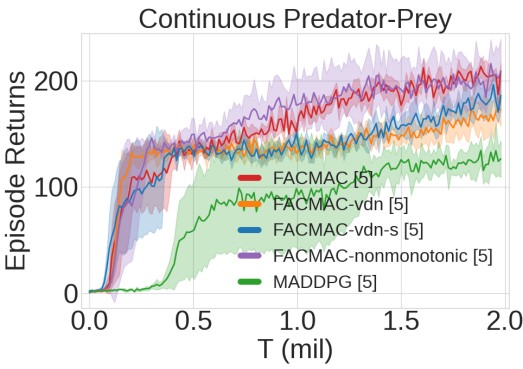

Figure 12: Mean episode return on our Continuous Predator-Prey task (with 3 agents and 1 prey).

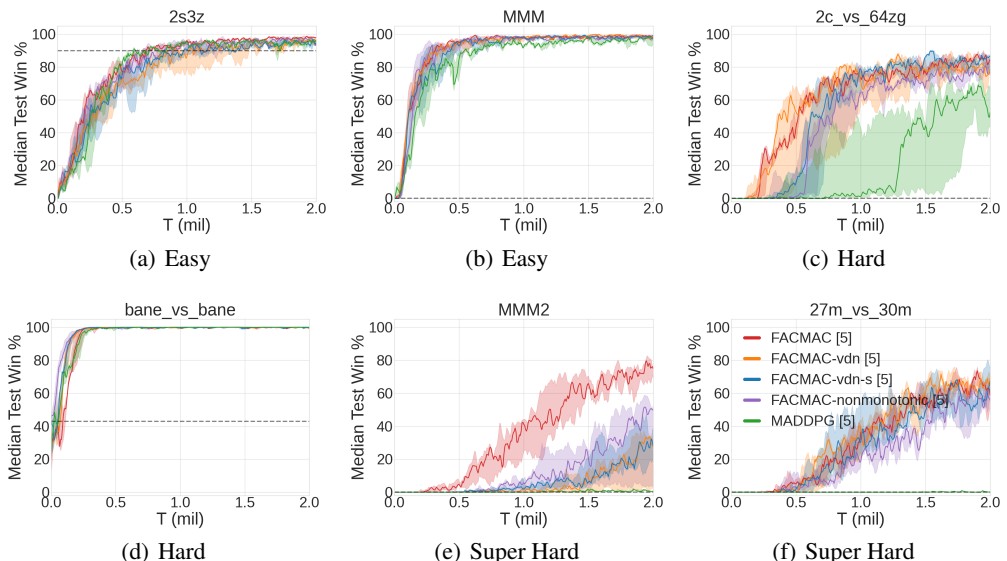

Figure 13: Median test win % on six different SMAC maps: (a) *2s3z* (easy), (b) *MMM* (easy), (c) *2c_vs_64zg* (hard), (d) *bane_vs_bane* (hard), (e) *MMM2* (super hard), and (f) *27m_vs_30m* (super hard), comparing FACMAC with different forms of critic factorisations.