# OpenReview forum: "FACMAC: Factored Multi-Agent Centralised Policy Gradients"
_NeurIPS.cc/2021/Conference — NeurIPS 2021 Poster_

### Official Review · Reviewer_X7Th · 2021-07-15

**Rating:** 6
**Confidence:** 4

**Summary:**

The paper proposes a centralized policy gradient actor-critic method for multiagent cooperation environments. The critic network follows the CTDE value-based training model, like VDN and QMix, but with flexible factorization forms. The actor network is updated by a centralized policy gradient. The proposed method is not just suitable for the continuous action space but also extends to the discrete action space. The method shows superior performance over other baselines in three domains, including a novel MAMujoco environment, which is first released with this paper.

**Main Review:**

Policy gradient methods have achieved success in the single-agent task, and promote it to multi-agent tasks is a promising and important direction. This paper takes advantage of two kinds of existed multi-agent methods and proposes a novel combination.  A multiagent benchmark MAMujocu is also released, which may as a good testbed for multi-agent research.

The paper is written well and easy to follow, related work and existing methods are well placed and cited.

The experiments are conducted in three domains both in continuous and discrete action space. The evaluations are fairly sufficient and encourage. I believe it could drive more research to further explore the policy gradient methods in multi-agent tasks.

Some weaknesses and questions:

1. Some experiment results in figures are not consistent with the original paper,  such as in the MMM2 map, DOP reaches about 50% win in the original paper but only 20% in this paper. I don't check all the plots so I can't guarantee there existed no more mismatches. The author should explain how the experimental data are acquired? Why not directly use the data from other related papers.

2. In my view, the whole method is essentially more like a centralized training method, if so, why not compare it with the centralized policy gradients methods, such as centralized DDPG or PPO? Does that may work?

3. The DOP method utilizes offline training to enhance the efficiency of samples,  and the Q value target is combined with the online q target and tree-backup q target. I know FACMAC follows the deterministic policy gradient and extends to discrete tasks via Gumbel-softmax trick, but I still wonder why it works so nicely? Are there some advanced tricks or more analysis?

4. Different critic factorization leads to inconsistent superiority in different tasks. I suggest the author make a deeper study and give more reasonable explanations.

**Time Spent Reviewing:**

5

---

> ### Author Response · Authors · 2021-08-10
> **Replying to Reviewer X7Th**
>
> Thank you for your review.
>
> “Some experiment results in figures are not consistent with the original paper, such as in the MMM2 map, DOP reaches about 50% win in the original paper but only 20% in this paper. I don't check all the plots so I can't guarantee there existed no more mismatches. The author should explain how the experimental data are acquired? Why not directly use the data from other related papers.” \
> \> Our DOP results were obtained by running DOP using the default open-sourced code (without any changes) released from the DOP paper [1]. One important thing to note is that we use SC2 version 4.10 in our experiments (as mentioned in footnote 4 in our paper), which is not directly comparable to results obtained using 4.6. [1] does not specify which version they use. Another difference is that we report the median test win rate as well as the 25-75% percentiles using 5 random seeds, while [1] reports the mean test win rate with 95% confidence interval using 12 random seeds. Thus, we do not expect our learning curves on SMAC to be exactly the same as the ones in [1].
>
> “In my view, the whole method is essentially more like a centralized training method, if so, why not compare it with the centralized policy gradients methods, such as centralized DDPG or PPO? Does that may work?” \
> \> FACMAC uses centralised training but *not* centralised execution. We only consider partially observable cooperative multi-agent tasks in this paper, which can be modeled as Dec-POMDPs. This means during execution, each agent must act with only access to its own local action-observation history. Thus, methods such as centralised DDPG or PPO cannot be used directly in our setting since they would require information sharing between agents during the execution phase. \
> Since our setting allows for centralised training we take advantage of that as much as possible, thus during the training phase our method does resemble DDPG. But importantly, we carefully design its architecture to ensure we can decentralise the agents.
>
> “The DOP method utilizes offline training to enhance the efficiency of samples, and the Q value target is combined with the online q target and tree-backup q target. I know FACMAC follows the deterministic policy gradient and extends to discrete tasks via Gumbel-softmax trick, but I still wonder why it works so nicely? Are there some advanced tricks or more analysis?” \
> \> FACMAC achieves competitive results on SMAC since 1) the centralised but factored critic allows it to scale to more complex tasks, 2) the proposed centralised policy gradient allows for better coordination among agents, and 3) it is off-policy and hence avoids problems related to non-stationarity when training on older data. \
> In particular, the ability to use off-policy data, in a similar manner to Q-learning, contributes to our sample efficiency significantly which can be seen when comparing the performance of DOP with FACMAC-vdn. \
> We do not use any other tricks other than the Straight-Through Gumbel-Softmax (as described in Lines 216-222) when adapting FACMAC to discrete action spaces. We did find that using Straight-Through Gumbel-Softmax tends to achieve better performance than using the Gumbel-Softmax trick. FACMAC is thus much simpler than stochastic DOP, which is another strength of our method.
>
> “Different critic factorization leads to inconsistent superiority in different tasks. I suggest the author make a deeper study and give more reasonable explanations.” \
> \> Thank you for the suggestion. We aim to investigate the benefits of different types of critic factorisations in different multi-agent tasks. We believe our experimental results provide some good reference points about what type of critic factorisation is expected to perform well in different cooperative tasks. One big takeaway from our results is that the factorisation we use in FACMAC is a consistently strong performer, and that factorisation as a whole is very important for achieving good performance. We will utilise the extra page allowed for accepted papers to provide a more detailed analysis about this.
>
> [1] DOP: Off-Policy Multi-Agent Decomposed Policy Gradients. Wang et al. ICLR 2021.

---

### Official Review · Reviewer_oCU2 · 2021-07-16

**Rating:** 6
**Confidence:** 4

**Summary:**

This paper proposed an actor-critic-based multi-agent deep reinforcement learning approach in a fully cooperative setting. Specifically, instead of using a monolithic centralized critic, the authors proposed to use a factorized critic, where the individual agent’s value function is nonlinear combined into a joint value function. In addition, the authors proposed a centralized policy gradient estimator where joint actions are sampled from current policies. The approach is evaluated on MPE, SMAC, and  multi-agent MuJoCo, which is a new multi-agent environment.

**Ethical Concerns:**

No, the reviewer doesn't see any ethical issues with this paper.

**Limitations And Societal Impact:**

Yes, the authors discuss the limitation and  societal impact of their work.

**Main Review:**

The paper applied the factorized value function, which is often used in value-based approaches such as QMIX, to actor-critic based methods such as MADDPG. It is impressive that the proposed approach achieves good experimental results on SMAC which was mostly dominated by value-based approaches. The paper is generally well written and easy to follow. However, the reviewer has some concerns about the motivation of the centralized gradient and the scalability of the proposed factorized critic. Please see the detailed comments and questions below.

1. As the motivation for the centralized gradient, the authors mentioned that assuming all other agent’s action is fixed while optimizing one agent’s policy will lead to suboptimal policies. The reviewer doesn’t see any intuitive or theoretical reason why this leads to suboptimal policies. Could you elaborate?

2. The authors claim that sample other agents’ action from the replay buffer leads to suboptimal policies. Again, it is unclear to the reviewer why this leads to suboptimal policies.

3. The authors claim that the factorized critic is more scalable. How does it compare with GCN-based centralized critic, such as  [1, 2]? [1, 2] scales to hundreds of agents in the MPE environment. In this paper, only nine agents are considered in the MPE environment.

[1] Graph Convolutional Reinforcement Learning, Jiang et. al. ICLR 2020

[2] PIC: Permutation Invariant Critic for Multi-Agent Deep Reinforcement Learning, Liu et. al. CoRL 2019




**Time Spent Reviewing:**

5 hours

---

> ### Author Response · Authors · 2021-08-10
> **Replying to Reviewer oCU2**
>
> Thank you for your review.
>
> “As the motivation for the centralized gradient, the authors mentioned that assuming all other agent’s action is fixed while optimizing one agent’s policy will lead to suboptimal policies. The reviewer doesn’t see any intuitive or theoretical reason why this leads to suboptimal policies. Could you elaborate?” \
> \> Figure 2(b) provides intuition for why the per-agent policy gradient can lead to sub-optimal policies even in a simple continuous matrix game. It shows that, when using per-agent policy gradient, the gradient for agent 1 at the origin is 0 (similarly for agent 2) since the gradient term assumes agent 2’s action to be fixed and thus it only considers the relative improvements along the dotted line (agent 1’s own action space). This will lead to the joint-action policy remaining at the origin if we computed and followed the exact gradients. In contrast, our centralised policy gradient is able to correctly determine the gradient for improving the joint action by optimising over the entire joint action space.
>
> “The authors claim that sample other agents’ action from the replay buffer leads to suboptimal policies. Again, it is unclear to the reviewer why this leads to suboptimal policies.” \
> \> When sampling the other agents’ actions from the replay buffer, their sampled actions might be drastically different from the actions their current policies would choose. This is because their policies are changing due to learning throughout training. Thus, we are effectively learning a best-response against a mixture of the past policies which can lead to the same pathologies as relative overgeneralization.
>
> “The authors claim that the factorized critic is more scalable. How does it compare with GCN-based centralized critic, such as [1, 2]? [1, 2] scales to hundreds of agents in the MPE environment. In this paper, only nine agents are considered in the MPE environment.” \
> \> The scalability of FACMAC is demonstrated by our experiments on SMAC maps 27m_vs_30m and bane_vs_bane, which feature 27 and 24 agents, respectively. Exploring the scalability of FACMAC compared to GCN-based centralised critic is an interesting direction for future work. \
> A benefit of our setup is that one could potentially utilise a GCN as part of its architecture. \
> Another important point to note is that the SMAC tasks we test on feature significantly more complex environmental dynamics than the MPE environment.

---

> > ### Comment · Reviewer_oCU2 · 2021-08-31
> > **Thanks for response**
> >
> > Thanks for the reply. The response addressed my concern. I encourage the author to incorporate the discussion on scalability into the update version.

---

> > > ### Author Response · Authors · 2021-08-31
> > > **Thank you**
> > >
> > > Thank you for your comments. We are pleased that our response addressed your concern. We will include the discussion on scalability in the revised version.

---

### Official Review · Reviewer_q18r · 2021-07-17

**Rating:** 7
**Confidence:** 3

**Summary:**

The paper proposed the FACMAC method, an actor-critic approach for the cooperative MARL tasks. Under the centralized training with decentralized execution setting, FACMAC learns a centralized but factored critic and updates the decentralized policy of each agent using a centralized gradient estimator. Comparing to the existing approaches, the proposed approach 1) enjoys the flexibility to use nonmonotonic factored critics to handle the tasks that cannot be solved with monolithic or monotonically factored critics; and 2) using the centralized policy gradients for decentralized policy training prevents being stuck in suboptimal solutions.

**Limitations And Societal Impact:**

The authors addressed them.

**Main Review:**

The main novelty of the work is to use a centralized but factored critic and a centralized gradient for training decentralized policies. Some of the ideas are brought from existing approaches, but the authors are able to explain and compare the differences clearly. The extensive empirical studies show the potential power of the proposed approach, and also validates the advantages claimed in previous sections. The Multi-Agent MuJoCo benchmark built for continuous cooperative MARL can be meaningful. Overall, the materials are presented in a clear and precise way.

Major Questions:
1. Can the authors comment on what kinds of tasks cannot be handled by FACMAC and may require a FACMAC-nonmonotonic approach?
2. As there're no restrictions on the factored critics, is it possible for the proposed algorithm to go beyond cooperative tasks?

**Time Spent Reviewing:**

4

---

> ### Author Response · Authors · 2021-08-10
> **Replying to Reviewer q18r**
>
> Thank you for your review.
>
> “Can the authors comment on what kinds of tasks cannot be handled by FACMAC and may require a FACMAC-nonmonotonic approach?” \
> \> FACMAC can perform poorly in tasks with *nonmonotonic* value functions, as demonstrated by our experimental results in Figure 8. FACMAC uses a QMIX-style factorisation. Hence the monotonicity constraint prevents it from representing joint action-value functions that are characterised as nonmonotonic [1], i.e., an agent’s ordering over its own actions depends on other agents’ actions (as discussed in Lines 552-554 in Appendix A). Therefore, in tasks that require significant coordination among agents within a given timestep, FACMAC-nonmonotonic can potentially perform better than FACMAC. [1, 2, 3] explore these kinds of tasks and situations where a QMIX-style factorisation can perform poorly. We will make this clearer.
>
> “As there're no restrictions on the factored critics, is it possible for the proposed algorithm to go beyond cooperative tasks?” \
> \> We think it is possible to adapt FACMAC to competitive or mixed settings. One possibility is to learn a *separate* centralised and factorted critic for each agent, such that agents can have arbitrary reward functions, including conflicting rewards in competitive tasks. It is unclear if the factored critic can help improve performance in these settings. We are optimistic it can, since we observe many benefits to factoring the critic in complex tasks. This can be an interesting avenue for future work.
>
> [1] MAVEN: Multi-Agent Variational Exploration. Mahajan et al. NeurIPS 2019. \
> [2] QTRAN: Learning to Factorize with Transformation for Cooperative Multi-Agent Reinforcement Learning. Son et al. ICML 2019. \
> [3] Weighted QMIX: Expanding Monotonic Value Function Factorisation for Deep Multi-Agent Reinforcement Learning. Rashid et al. NeurIPS 2020.

---

### Official Review · Reviewer_ZyWj · 2021-07-17

**Rating:** 4
**Confidence:** 3

**Summary:**

This paper presents a centralized multi-agent actor-critic method with a factored critic, which combines per-agent utilities (Q values) into a mixing function, as in QMIX. The difference is that it removes the constraint of monotonicity (using non-negative weights in the mixer) for higher representational capacity. It also proposes a centralized policy gradient estimator to fully take the advantage of the centralized critic, which updates policies over joint action space instead of optimising each agent's policy separately. Along with the FACMAC algorithm, Multi-agent MuJoCo, a new multi-agent benchmark suite, is introduced for cooperatively tasks with continuous control based on single-agent MuJoCo.

**Main Review:**

Generally, FACMAC is similar to a combination of MADDPG and QMIX in structure, e.g. the QMIX with actors or the MADDPG with a mixer. But this combination is valuable from the experimental results.

The author claims there are no monotonic constraints in FACMAC but no directly theoretical proof is found in the paper, which is said to be important for maintaining consistency between policies in QMIX. Is this because of the use of Centralised Policy Gradients? If so, rigorous proof will be appreciated.

In terms of Centralised Policy Gradients, the author has not provided the theoretical proof for why optimising joint policy is better than optimising each agent’s policy separately as well.

The Multi-agent MuJoCo is quite useful when I trying to test it with some benchmark algorithms, all agents have a clear division of labor and their cooperation is well emphasized.

In the experiment, there are many classic scenarios in MuJoCo but only three of them are selected to be tested. If more scenarios are tested, the results on continuous control will be more persuasive.

Pros:
Critic factorisation is more flexible without monotonic constraints.
The Multi-agent MuJoCo is a practical suite.
The Paper is overall well-written and clear.

Cons:
Lack of theoretical proof. e.g. Why monotonic constraints could be removed? Why do Centralised Policy Gradients help?
Novelty may not be enough. What's the key difference comparing to QMIX with actors/MADDPG with a mixer?

**Time Spent Reviewing:**

4

---

> ### Author Response · Authors · 2021-08-10
> **Replying to Reviewer ZyWj**
>
> Thank you for your review.
>
> “The author claims there are no monotonic constraints in FACMAC but no directly theoretical proof is found in the paper, which is said to be important for maintaining consistency between policies in QMIX. Is this because of the use of Centralised Policy Gradients? If so, rigorous proof will be appreciated.” \
> \> There are no constraints on factoring the critic in FACMAC as the critic is not used for greedy action selection in actor-critic methods. Like all actor-critic methods, DDPG in particular, there are no guarantees that the actor’s actions are exactly maximising the critic. However, in an actor-critic setup this is absolutely fine and expected. In contrast, QMIX would not work sensibly without this consistency.
>
> “In terms of Centralised Policy Gradients, the author has not provided the theoretical proof for why optimising joint policy is better than optimising each agent’s policy separately as well.” \
> \> Figure 2(b) provides a specific example of when our centralised policy gradient is better than per-agent policy gradient. It shows that the per-agent policy gradient can be wrong even in a simple continuous matrix game, due to assuming the other agent’s action to be fixed. However, our centralised policy gradient is able to correctly determine the gradient for improving the joint action by optimising over the entire joint action space.
>
> Additionally, we have sufficient empirical evidence demonstrating the benefit of our centralised policy gradient over per-agent policy gradient on three different domains (continuous matrix game, SMAC, and MAMuJoCo), as shown in Figures 2(a) and 7. These results show that both MADDPG and FACMAC yield significant performance improvements when using our centralised policy gradient (CPG). More specifically, MADDPG (with CPG) performs significantly better than MADDPG on both continuous matrix game and SMAC map 2c_vs_64zg. FACMAC performs significantly better than FACMAC (without CPG) on both SMAC map MMM2 and ManyAgent Swimmer [10x2] on MAMuJoCo, and has lower variance across seeds on SMAC map 2c_vs_64zg.
>
> “The Multi-agent MuJoCo is quite useful when I trying to test it with some benchmark algorithms, all agents have a clear division of labor and their cooperation is well emphasized.” \
> \> We are pleased that you like our new MAMuJoCo benchmark. We believe it can be very useful for the MARL community, e.g., stimulate more progress in continuous MARL.
>
> “In the experiment, there are many classic scenarios in MuJoCo but only three of them are selected to be tested. If more scenarios are tested, the results on continuous control will be more persuasive.” \
> \> We have run experiments on more MAMuJoCo scenarios including 2-Agent HalfCheetah and 2-Agent Walker. The same conclusion holds. FACMAC is still one of the best performing methods in both scenarios. We will include these results in a future revision, but cannot update the paper with these new experiments during the review period.
>
> “Cons: Lack of theoretical proof. e.g. Why monotonic constraints could be removed? Why do Centralised Policy Gradients help? Novelty may not be enough. What's the key difference comparing to QMIX with actors/MADDPG with a mixer?” \
> \> We disagree that the lack of theoretical proof is a weakness of the paper. Empirical research is a legitimate form of contribution. The new algorithms, benchmark, and empirical findings will all be very useful to the MARL community. \
> As mentioned above, the monotonic constraints on the critic can be removed since we are utilising an actor-critic setup. There are no theoretical results that need to be shown, proven or justified when removing the monotonicity constraints in the critic. We have also provided a clear example of when our centralised policy gradient estimator can succeed when the per-agent policy gradient estimator can fail. \
> Furthermore, the lack of formal theoretical guarantees in the form of regret bounds for instance is typical in deep RL, e.g., QMIX does not have any such theoretical guarantees.
>
> One crucial difference between FACMAC and ‘MADDPG with a mixer’ is the use of our new centralised policy gradient, which is demonstrated to be able to fully reap the benefits of a centralised critic and to allow for more coordinated policy changes. \
> A crucial difference to QMIX is that our framework allows for a more flexible factorisation of the joint-action Q. We have explored a new nonmonotonic factorisation and demonstrated its advantages in some tasks as shown in Figure 8 to take advantage of this. \
> Furthermore, QMIX is a Q-learning algorithm that does not fit into an actor-critic framework.

---

### Official Review · Reviewer_JS3e · 2021-07-24

**Rating:** 7
**Confidence:** 1

**Summary:**

In this paper, the authors propose a FACtored Multi-Agent Centralised PG (FACMAC) method to solve the fully cooperative MARL problems.  The proposed method is under the paradigm of centralized training with decentralized execution.  The critic of the proposed method is centralized and factored, in the sense that the joint action-value function is a combination of the utility of each agent. A new benchmark MAMuJoCo (multi-agent variant of MuJoCo) is proposed to measure the performance of FACMAC. And the proposed method behaves the best compared with several other popular MARL algorithms.




**Ethical Concerns:**

No ethical concern.

**Limitations And Societal Impact:**

This work is an algorithm that solves certain high-level abstract RL problems, it does not focus on any specific problems that may have negative societal impacts.

**Main Review:**

The reviewer is not an expert in this area. From the reviewer's point of view, the paper combines the technique of several MARL algorithms such as MADDPG and QMIX. The novelty is that compared to MADDPG which optimizes each agent separately while fixing other agents, the FACMAC directly optimizes the joint action. Compared to QMIX, FACMAC allows the user to apply richer value factorizations, which can be nonmonotonic. Because this work is not a theoretical study that focuses on the sample complexity of a certain algorithm, the practical performance of the proposed framework will be the main judging criteria. From this perspective, FACMAC shows advantages in various tested benchmark problems against the compared algorithms. Therefore, the reviewer considers this as a good submission.

**Time Spent Reviewing:**

8

---

> ### Author Response · Authors · 2021-08-10
> **Replying to Reviewer JS3e**
>
> Thank you for your review. We are pleased that you think it is a good submission.

---

### Decision · Program_Chairs · 2021-09-28

**Decision:**

Accept (Poster)

**Comment:**

FACMAC uses the ideas of MADDPG and QMIX, where its structure has both a mixer and an actor. The proposed algorithm was empirical tested on MPE, SMAC, and multi-agent MuJoCo, and demonstrated superior performances. Reviewers appreciate the algorithm's empirical performances, especially because policy gradient-type methods didn't work well on such tasks (multi-agent mojuco) prior to this submission. Although the algorithmic idea is rather natural, the empirical results are strong for multi-agent cooperative RL.

One remaining comment is that the paper is purely empirical and lacks theoretical proof to justify central policy gradient and policy consistency without monotonicity constraint. Although theoretical proof is not a must, we strongly recommend that the authors provide more theoretical justifications and explanations, so that people can learn useful insight from the pure empirical work. Another remaining comment is the method relies on centralized training and only the execution is decentralized. ers

**Consistency Experiment:**

NeurIPS has a long history of experimentation. In 2014, NeurIPS ran an experiment in which 10% of submissions were reviewed by two independent committees to quantify the randomness in the review process. This year, we repeated a variant of this experiment to see how the quality of the review process has changed over time.  This paper was part of the experiment and was therefore assigned to two committees (consisting of reviewers, an Area Chair, and a Senior Area Chair) that reached independent decisions.  If both committees made the same recommendation, this recommendation was followed. If a single committee recommended acceptance, the paper was accepted (with the exception of a few cases in which the other committee identified what we considered a fatal flaw, e.g., an error in a key result).

This copy’s committee reached the following decision: **Accept (Poster)**

The other committee assigned to the paper recommended **Reject**.  You can find the other set of reviews, along with any follow up discussion with the authors here:
https://openreview.net/forum?id=wZYWwJvkneF